



# Contrasted release of insoluble elements (Fe, Al, REE, Th, Pa) after dust deposition in seawater: a tank experiment approach

Matthieu Roy-Barman [1], Lorna Folio[1], Eric Douville[1], Nathalie Leblond[2], Frédéric Gazeau[3], Matthieu Bressac[3,4], Thibaut Wagener[5], Céline Ridame[6], Karine Desboeufs[7], Cécile Guieu[3]

5    [1] Université Paris-Saclay, CNRS, CEA, UVSQ, Laboratoire des Sciences du Climat et de l'Environnement, 91191, Gif-sur-Yvette, France

[2] Sorbonne Université, CNRS, Institut de la Mer de Villefranche, IMEV, F-06230 Villefranche-sur-Mer, France

[3] Sorbonne Université, CNRS, Laboratoire d'Océanographie de Villefranche, LOV, F-06230 Villefranche-sur-Mer, France

10   [4] Institute for Marine and Antarctic Studies, University of Tasmania, Hobart, Tasmania, Australia

[5] Aix Marseille Univ., Université de Toulon, CNRS, IRD, MIO UM 110, 13288, Marseille, France

[6] Sorbonne Université, LOCEAN, 4 Place Jussieu – 75252 Paris Cedex 05, France

[7] LISA, UMR7583, Université de Paris, Université Paris-Est-Créteil, Institut Pierre Simon Laplace (IPSL), Créteil, France

*Correspondence to*: Matthieu Roy-Barman (matthieu.roy-barman@lsce.ipsl.fr)

## Submitted to Biogeosciences (July 28th, 2020)



**Abstract.**

The release of lithogenic elements (which are often assumed to be insoluble) such as Aluminum (Al), Iron (Fe), Rare Earth Elements (REE), Thorium (Th) and Protactinium (Pa) by Saharan dust reaching Mediterranean seawater was

5    studied through tank experiments over 3 to 4 days under controlled conditions including control without dust addition and dust seeding under present and future climate conditions (+3 °C and -0.3 pH unit). Unfiltered surface seawater from 3 oligotrophic regions (Tyrrhenian Sea, Ionian Sea and Algerian Basin) were used. The maximum dissolution fractions were low for all seeding experiments: less than 0.3% for Fe, 1% for $^{232}$Th and Al, about 2-5% for REE and less than 6% for Pa. Different behaviors were observed: dissolved Al increased until the end of the experiments, Fe

10   did not dissolve significantly and Th and light REE were scavenged back on the particles after a fast initial release. The constant $^{230}$Th/$^{232}$Th ratio during the scavenging phase suggests that there is little or no further dissolution after the initial Th release. Quite unexpectedly, comparison of present and future conditions indicates that changes in temperature and/or pH influence the release of thorium and REE in seawater, leading to a lower Th release and a higher light REE release under increased greenhouse conditions.



## 1 Introduction

The ocean biological productivity is strongly controlled by inputs of trace metals like iron (Fe), a limiting micronutrient for marine primary producers. Aeolian dust deposition over the ocean represents a significant Fe source for marine
surface ecosystems (Duce and Tindale, 1991, Jickells et al., 2005). However, the aeolian Fe flux is difficult to quantify precisely, because the aeolian dust deposition flux, the solubility of Fe and the removal rate of dissolved Fe (by biotic and/or abiotic processes) are poorly constrained (Baker and Croot, 2010). To disentangle these processes, lithogenic tracers such as Aluminum (Al), Thorium (Th), Protactinium (Pa) and Rare Earth Elements (REE) that are not involved (or not as much as Fe) in biological processes are used to determine the inputs of lithogenic dust (Measures and Vink,
2000, Hsei et al., 2011, Greaves et al., 1999). This is based on the premise that the content of these lithogenic tracers in surface waters should be more or less proportional to their release rate by dissolution of aeolian dust as long as they are not actively removed by the biological activity. Moreover, thorium has one isotope ($^{232}$Th) derived from the lithogenic material dissolution, whereas another one ($^{230}$Th) is mostly produced in seawater by radioactive decay of conservative $^{238}$U and hence can be used as a chronometer of the input and removal rate of $^{232}$Th in ocean surface
waters. A key, but poorly constrained parameter used in these methods is the solubility of these lithogenic elements (Anderson et al., 2016).

Here, we simulated Saharan dust deposition in surface Mediterranean seawater to determine the release of selected lithogenic tracers (Fe, Al, REE, Th and protactinium (Pa)). The main objective is to determine the (absolute or at least relative) solubility of these tracers, their dissolution kinetics and the possible influence of temperature, pH and
biological activity.

## 2 Methods

### 2.1 Experimental setup

A detailed description of the artificial dust addition experiments is given in Gazeau et al. (2020a, this issue). In brief, six experimental High Density PolyEthylene (HDPE) tanks (300 L each, with a conical base connected to a
sediment trap) in which the irradiance spectrum and intensity can be finely controlled and in which future ocean acidification and warming conditions can be fully reproduced, were installed in a temperature-controlled container during the PEACETIME cruise (doi: 10.17600/17000300). This cruise was conducted on board the R/V Pourquoi Pas? in the Mediterranean Sea during the season characterized by strong stratification (Guieu et al., 2020, this issue). Three stations covering different *in situ* conditions but all characterized by oligotrophic conditions were chosen to conduct
tank experiments of 72 h (3 days): stations TYR in the Tyrrhenian Sea, ION in the Ionian Sea and FAST in the Algerian basin (Fig. ES1, Guieu et al., 2020, this issue). The last experiment at station FAST was extended to four days. The experimental tanks were filled with unfiltered seawater from the continuous surface (5 m) pumping system upon arrival at stations TYR (17/05/2017) and ION (25/05/2017) and one day after arrival at station FAST (02/06/2017). Tanks C1 and C2 were unmodified control tanks, D1 and D2 were enriched with dust (3.6 g of dust per tank which correspond
to a realistic dust input of 10 g m$^{-2}$), and G1 and G2 were warmed (+3 °C), acidified (-0.3 pH unit) and enriched with dust (same flux than in D1 and D2). The atmosphere above tanks C1, C2, D1 and D2 was flushed with ambient air and tanks G1 and G2 were flushed with air enriched with $CO_2$ (at 1000 ppm) in order to prevent $CO_2$ degassing from the





acidified tanks. The originality of this device is that the height of the tank (1.1 m) allows to take into account the settling of the particles and to analyze a series of parameters both in the tank and at its base (sediment trap).

Dust particles were derived from the fine fraction (< 20 µm) of a Saharan soil (Tunisia) and processed physically and chemically (including a treatment simulating the effect of cloud water) to produce an analogue of

Saharan dust deposited over the Mediterranean Sea (see details in Guieu et al., 2010). The size spectrum of these dust presents a median diameter around 6.5 µm and a peak at ~ 10 µm similar to the one found in Mediterranean aerosols (Guieu et al., 2010). It is a mixture of carbonates, quartz and clays. Chemically, it contains 3.3 % of Al and 2.3 % of Fe (Desboeufs et al., 2014). The detailed dust seeding procedure is given in Gazeau et al. (2020, this issue). All tanks were sampled for dissolved Fe and Al before dust enrichment (t = 0 h) and then, 24 h and 72 h (TYR and ION) or 96

h (FAST) after dust enrichment. Samples for Rare Earth Elements (REE), Th and Pa were not taken at station TYR. At station ION, all tanks were sampled for Al, Fe and REE at t = 1, 24 and 72 h. At station FAST, tanks C1 and D1 were sampled for Al, Fe and REE at t = 0, 1, 6, 12, 24, 48, 72 and 96 h. Tanks G1 and G2 were sampled at t = 1, 48 and 96 h after dust enrichment.

At the end of each experiment, the particulate material that settled at the bottom of the tanks was recovered

from the sediment traps and preserved by adding formaldehyde (final concentration 5%).

### 2.2 Analytical techniques

#### 2.2.1 Dissolved Fe

Dissolved iron (DFe) concentrations were measured by flow injection with online preconcentration and

chemiluminescence detection (Bonnet and Guieu, 2006; Guieu et al. 2018). An internal acidified seawater standard was measured daily to control the stability of the analysis. During the analysis of the samples collected during the PEACETIME cruise, the detection limit was 15 pM and the accuracy of the method was controlled by analyzing the SAFe S ($0.086 \pm 0.010$ nM (n = 3); consensus value $0.093 \pm 0.008$ nmol·kg$^{-1}$, SAFe D1 ($0.64 \pm 0.13$ nmol·kg$^{-1}$ (n = 19); consensus value $0.67 \pm 0.04$ nM), GD ($1.04 \pm 0.10$ nM (n = 10); consensus value $1.00 \pm 0.10$ nmol·kg$^{-1}$), and GSC

($1.37 \pm 0.16$ nmol·kg$^{-1}$ (n = 4); consensus value not available) seawater standards.

#### 2.2.2 Dissolved Al

Determinations of dissolved aluminum (DAl) concentrations were conducted on board using the fluorometric

method described by Hydes and Liss (1976). After filtration, samples were acidified to pH < 2 with double distilled concentrated HCl. After at least 24 h, the lumogallion reagent was added to the sample, which was then buffered to pH 5 with ammonium-acetate. The sample was then heated to 80°C for 1.5 h to accelerate the complex formation. The fluorescence of the sample was measured with a Jasco FP 2020 + spectrofluorometer (excitation wavelength 495 nm, emission wavelength 565 nm). The detection limit varied between 0.2 and 0.5 nM and the blank values between 0.9

and 1.7 nM for the different days of analysis. Based on the daily analysis of an internal reference standard seawater, the overall repeatability of the method was 0.6 nM (standard deviation on a mean concentration of 53.5 nM, n = 25).

#### 2.2.3 Dissolved REE, Th and Pa

Seawater was sampled from the tanks and filtered (pore size 0.45/0.2 µm; Sartobran®) within 1-2 h after sampling. Seawater was then acidified with trace metal grade HCl (NORMATOM®). Approximately 250 mL of filtered seawater was spiked with isotopes $^{150}$Nd, $^{172}$Yb, $^{229}$Th and $^{233}$Pa for isotope dilution measurements and KMnO$_4$ and MnCl$_2$ were added. Then REE, Th and Pa were pre-concentrated by co-precipitation of MnO$_2$ obtained by raising



pH to 8 through addition of concentrated $NH_3$ The $MnO_2$ precipitate was then recovered by filtration onto a 25 mm cellulose ester filter, rinsed with MQ water and dissolved in a solution composed of 2 mL of 6N HCl and 10 µL of $H_2O_2$. REE, Th and Pa were then separated using an AG1X8 ion exchange column (Gdaniec et al., 2018).

REE contents were measured at the LSCE by using a quadrupole ICPMS (Xseries[II], Thermo ScientificⒸ).
Nd and Yb concentrations were directly determined by isotope dilution. Comparison of these ID-concentrations with the concentration determined by internal calibration (using In-Re internal standard) provided the yield of the chemical procedure for Nd and Yb (~70-100%). These two chemical yields were then used to estimate the yields of the other REE, by assuming that within the REE group, this yield is a linear function of the atomic number (Arraes-Mescoff et al., 1998).

Pa and Th analyses were performed using an Inductively Coupled Plasma Mass Spectrometer (MCICP-MS, Neptune[plus] Ⓒ) equipped with a Secondary Electron Multiplier (SEM) and a Retarding Potential Quadrupole (RPQ) energy filter (Gdaniec et al., 2018).

Analyses of seawater used during the GEOTRACES intercalibration exercise (van der Fliert et al., 2012) generally showed agreements within a few percents with consensual values except La and Lu that were as underestimated by 25% and 10% (Tab. ES1). Agreement within analytical uncertainties were obtained for $^{232}Th$ and age-corrected $^{230}Th$ concentrations. The very large uncertainties on 232Th analyses of the GEOTRACES standard were due to its low $^{232}Th$ content (particularly compared to Mediterranean seawater and the small sample volume used. 231Pa values are not reported because they correspond to the analysis series where yield and Blank problems were encountered for 231Pa analysis (see section 3.5).

### 2.2.4 Trapped particles

Samples were treated following the standard protocol developed at the national service "Cellule Piege" of the French INSU-CNRS (Guieu et al., 2005). Trapped particles were then rinsed three times with ultrapure (MilliQ) water in order to remove salt and then freeze-dried. Approximately 10 mg of particles were then weighed and $HNO_3/HF$ acid-digested using Suprapur reagents at 150 °C in PTFE vials. After complete evaporation, samples were diluted in 0.1 M $HNO_3$ and analyzed for Fe and Al concentrations by ICP-AES (JY 138"Ultrace", Jobin YvonⒸ). A fraction of the remaining solution was used to analyze REE, Th and Pa. For Th and Pa, the solution was spiked with $^{229}Th$ and $^{233}Pa$ and treated through the same chemical process as the Mn precipitate used for the dissolved Pa and Th analysis. REE were analyzed directly on a quadrupole ICPMS (Xseries[II], Thermo ScientificⒸ) using an internal calibration (Re).

## 3 Results

### 3.1 Dissolved Fe

Over the course of the three experiments, DFe concentrations in control tanks were in the range of 0.7-2.5 nM (Tab. ES2, Fig. 1) in good agreement with surface waters (0-15 m) DFe measured during the cruise (TYR: 1.47 ± 0.30 nM; ION: 1.41 ± 0.19 nM; FAST: 1.71 ± 0.35 nM, Bressac et al., in prep.) and more generally with surface concentrations observed in the Mediterranean Sea during the stratification period (Bonnet and Guieu 2006; Gerringa et al., 2017; Wagener et al., 2008). For the TYR experiment, there was no clear systematic difference between controls (C1 and C2) and dust amended tanks (D1, D2, G1 and G2) that would indicate a significant release of Fe from dust. For the ION experiment, the concentrations measured in G1 were much higher than in the other tanks and most likely highlighted a contamination issue. For the FAST experiment, DFe concentrations were lower in control tanks than in dust amended tanks. However, during this experiment, high variability between duplicates suggest possible contamination issues during sampling or sample processing.





### 3.2 Dissolved Al

The Al concentrations in control tanks varied between stations: ~ 50 nM at TYR, ~ 75 nM at ION and ~20-25 nM at FAST (Tab. ES2, Fig. 1). All these values are within the range of concentrations observed in Mediterranean surface waters (Rolison et al., 2015). At all three stations, Al concentrations measured before dust addition (t = 0 h)

were similar in all tanks. After dust addition, Al concentrations steadily increased in tanks D and G to reach final concentrations 50-100 nM higher than in control tanks with no systematic differences between D and G treatments. The concentration increase at FAST (72-80 nM) was higher than at TYR and ION (52-68 nM), due to a longer experiment at FAST.

### 3.3 Dissolved Rare Earth Elements

The REE concentrations measured in control tanks at stations ION and FAST (Tab. ES3, Fig. 2) compares well with values reported in the Mediterranean Sea (Censi et al., 2004, Tachikawa et al., 2008). In control tanks at both ION and FAST, there was a slight increase in concentrations during the course of the experiments suggesting some contamination from the tank or the environment of the experiment. However, this increase remained limited (i.e. ≈

+15-40% for dissolved Nd, ≈ +5-10% for dissolved Yb) compared to changes in concentrations observed in the dust amended tanks. For both D and G, there was a sharp increase in the concentrations of all REE (i.e. ≈ +400% for dissolved Nd, ≈ +100% for dissolved Yb), followed by a slow decrease. This decrease was steeper for light Rare Earth elements (LREE, e.g. Nd for which the concentration decrease was visible as early as t = +6 h) than for heavy Rare Earth elements (HREE, e.g. Yb for which the concentration remained relatively constant after t = +1 h). The only

exception in these regular trends was observed at FAST for tank D2, where no increase in REE concentrations was observed after dust addition (t = +1 h). As this most likely resulted from a technical issue during sampling (perhaps bottle labelling), we will consider this value as an outlier. In general, REE concentrations at a given time and site were higher (LREE) or equivalent (HREEE) in the warmer and acidified tanks (G) as compared to ambient environmental conditions (D).

### 3.4 Dissolved Thorium isotopes

Most $^{232}$Th concentrations in control tanks were about 1 pM both at FAST and ION (Tab. ES3, Fig. 3), in agreement with surface water concentrations in the Mediterranean Sea (Gdaniec et al., 2018). Only one high $^{232}$Th concentration was measured in tank C1 at station FAST 12 h after dust addition (~10 pM). Since the following value

measured in this tank (t = +24 h) was in the expected range (~1 pM), this extreme value likely resulted from a sample contamination, rather than a contamination of the tank itself. Slightly higher LREE concentrations for this sample, as compared to levels measured at the other time points supported this sample contamination hypothesis. REE have a longer residence time in seawater than Th and therefore, they are theoretically less sensitive to contamination from lithogenic material. As for REE, there was a sharp increase of $^{232}$Th concentrations after dust addition at both ION and

FAST. At FAST, concentrations were higher in tank D1 at t = +12 h and t = +24 h than at t = 1 h. However, as described above for tank C1 at station FAST and for the sampling time t = +12 h, we consider that these high values can be attributed to sample contamination during sampling. Therefore, we will not consider further these two samples (Fig. 3).

After the rapid $^{232}$Th increase for the D and G treatments at FAST and ION, there was a systematic decrease of the $^{232}$Th concentration. In contrast to REE, $^{232}$Th concentrations were higher in D tanks compared to G tanks.



The variations of $^{230}$Th concentrations with time and between treatments were similar than what described for $^{232}$Th concentrations. However, significant variations of the $^{230}$Th/$^{232}$Th ratio were observed (Tab. ES3, Fig. 3). The highest ratios ($^{230}$Th/$^{232}$Th $\geq 1 \times 10^{-6}$ mol/mol) were measured in controls, whereas lower ratios ($^{230}$Th/$^{232}$Th $\leq 1 \times 10^{-6}$ mol/mol) were found in D and G tanks.

### 3.5 Dissolved Protactinium

Due to analytical problems (low yield and large blanks) largely due to the small volumes available and low Pa content in the Mediterranean surface water, only Pa results obtained at FAST for tanks C1 and D1 will be presented. The mean $^{231}$Pa concentrations at FAST were not different within uncertainties in the C1 (2.5 ± 0.2 aM, with 1 aM = $10^{-18}$ M) and D1 (2.4 ± 0.2 aM) treatments (Tab. ES3, Fig. 4). Despite the small volumes of seawater analyzed, these concentrations agree within uncertainties, with the Pa concentrations available in surface western Mediterranean Sea (Gdaniec et al., 2017). Due to relatively large uncertainties on individual data, no systematic temporal trend is depictable.

### 3.6 Trapped material

The material collected in the traps contained 2.6% of Fe and 4.8% of Al (Tab. ES4). This is higher than the initial dust composition (2.3% of Fe and 3.3% of Al), due to preferential dissolution of calcium carbonate (Desboeufs et al., 2014). The lower Ca (14.2%) content in the trapped material compared to the initial dust (16.54%) suggests ~ 15% of calcium carbonate dissolution. REE concentrations in the sediment trap are close to concentrations in the average upper continental crust (Taylor and McLennan, 1995), yielding flat REE patterns (Tab. ES5, Fig. ES2). The particulate $^{232}$Th concentrations are within a range of 70 ± 5% of the upper continental crust concentration. The $^{230}$Th concentrations correspond roughly to secular equilibrium for a U/Th ratio of 0.40, in agreement with the range observed in the average continental crust. The $^{231}$Pa concentrations correspond to secular equilibrium for a U/Th ratio of 0.34, slightly below the ratio calculated with $^{230}$Th but still in agreement with the crustal range and Saharan aerosols (Pham et al., 2005).

### 4 Discussion

The concentration changes observed during the experiments resulted from a net balance between the release of chemical elements by the dissolution of the dust and removal of these elements on the particles by scavenging or active (biological) uptake. For an element like Fe, the scavenging efficiency largely depends on Fe stabilization in the dissolved phase by Fe-binding molecules (Witter et al. 2000). As the dust concentration was high, Fe readsorption on the dust could have been particularly high (Wagener et al., 2010). Dust inputs over the Mediterranean Sea are very irregular. The dust flux used for the seeding (10 g of total dust/m$^2$ with an Al content of 4%) corresponds to the highest dust pulses observed over the Mediterranean Sea and represents 30-100% of the yearly dust flux over the Mediterranean Sea (Guieu et al., 2010b). Hence, the PEACETIME experiments gives an idea of the yearly release of insoluble elements in the Mediterranean surface waters. To use the PEACETIME results in ocean basin with lower dust inputs, we will also evaluate the percentage of dissolution of these elements from the dust by dividing the observed concentration changes in the dissolved phase by the concentration of particulate element carried by dust in the tank (see equation 1 below).





### 4.1 Solubility of tracers

The percentage of dissolution of the different elements was calculated as the maximum release of the considered dissolved element during the experiments (largest difference in concentrations between D or G tanks and control tanks) divided by the amount of particulate tracers introduced in the tanks by dust addition (Tab. 1) following the equation:

$$f_{dissol\_conc} = \frac{CONC_{max} - CONC_{init}}{CONC_{dust} \times m/V} \times 100 \qquad\qquad (1)$$

10     .

where $CONC_{dust}$ is the concentration in the original dust (expressed in mol of insoluble element/g of dust). Direct analysis of the original dust was used for total Fe and Al (Guieu et al., 2010). For REE, Th and Pa, we used the average concentrations of particles collected in the traps decreased by 15% to account for the carbonate dissolution (see section 3.6). The mass of dust deposited in each tank (3.6 g) is noted m and V is the volume of the tank (300 L). During the

course of the experiments, the dust loss by sedimentation ranged from 28 to 65% likely depending on the intensity of aggregation in each tank (as previously observed by Bressac et al., 2011). However, it did not seemed to impact the estimation of the percentage of dissolution. For example, at the ION station, while a large difference was observed between the Al amount collected in the sediment traps of D1 and D2 (75% and 33% of the Al introduced when seeding were recovered in the traps, respectively, so that only 25% and 67% of the initial particulate remained in suspension

at the end of the experiment), the percentages of dissolution were identical in D1 and D2 for all the studied elements (Fig 1-3, Table 1).

    We suggested in section 3 that the DFe concentrations can be biased by contaminations during the

experiments. Nevertheless, we can put an upper limit to Fe dissolution by assuming that the highest DFe concentrations measured during the experiments truly represents Fe dust release. The highest DFe (10 nM) was measured at station FAST in the dust amended tank D1 at t = 72 h. Considering that when seeding the dust at the seawater surface of each tank, 30 μM of particulate Fe ($CONC_{dust}$ for Fe) were added, it follows that Fe dissolution extent is at most ~ 0.3%. This result is in good agreement with the percentage of dissolution of Fe obtained using the same dust and device with

filtered seawater from coastal Northwestern Mediterranean Sea under abiotic conditions in May (Bressac and Guieu, 2013, Louis et al., 2018).

For [231]Pa, we were not able to detect a significant difference between C1 and D1 at FAST. However, we can set an upper limit on Pa dissolution. Based on trap analyses (Tab. ES5), we estimate that $^{231}Pa_{dust} \times m/V = 0.04$ fM. Given the analytical uncertainties on dissolved [231]Pa analysis (Fig. 4), $CONC_{max} - CONC_{init}$ is certainly below 0.002 fM. Hence,

the percentage of dissolution of [231]Pa is below 5%. As expected for these poorly soluble elements, the maximum percentages of dissolution were low for all stations: less than 0.3% for Fe, 1% for [232]Th and Al (although their dissolution kinetics do not have the same patterns) and about 2-5% for REE.

Looking more into details, it appeared that Al dissolution was slightly higher at FAST compared to ION, but identical

in D and G treatments. The contrasting behaviors of Al that progressively dissolved during the experiments compared to Fe that did not dissolve significantly may be due to their respective solubility limit. The Al concentrations (~50 - 100 nM) during the experiments were much lower than the dissolved Al concentration in seawater at equilibrium with Al hydroxides which is at the micromolar level (Savenko and Savenko, 2011). By contrast, dissolved Fe concentrations



during the experiments were close or above the theoretical solubility of Fe hydroxides in seawater (Millero, 1998) due to the presence of Fe–binding ligands that keep Fe in solution (Wagener et al., 2008). For Al (as well as for Fe), there was no sample analysis at t = 1 h (just after dust addition), so it was not possible to detect a putative early dissolution as observed for REE and Th (see below).

For Th, the 2 samples considered as contaminated in section 3.4 are not considered hereafter (Fig. 3). Moreover, taking these two samples into account or not would not significantly affect the main conclusions of this study regarding Th. In contrast to Al, both Th and REE were released rapidly after dust addition, similarly to phosphate and nitrate (Gazeau et al., 2020, this issue). Fortuitously or not, it appears that more Th and DIP were released at FAST than at ION. Among the differences between ION and FAST, we note that FAST has a higher biomass than ION (although this is most visible at the end of the experiment, whereas Th release occurs during the first hour) and a lower alkalinity (Gazeau et al., 2020). However, this relationship with biomass does not hold if we compare the D and the G experiments. For both FAST and ION, there is more Th dissolution in the D tank compared to G tank (Fig. 2), whereas more biomass increase was observed under greenhouse conditions compared to normal conditions (Gazeau et al., 2020, this issue). The higher temperature imposed in tanks G induced a higher concentration of Transparent Exopolymeric Particles (TEP; Gazeau et al., in prep). The high affinity of Th for TEP (Santchi et al., 2004) combined with the aggregation of TEP on particles could limit the presence of Th in the G treatment as compared to D. At present, we must recognize that it is not possible to determine unambiguously the parameter(s) controlling the low percentage of Th release.

For REE, percentages of dissolution were relatively similar at FAST and ION, but unlike Th, slightly lower LREE dissolution fractions occurred under ambient environmental conditions (D, 1.9% for La) than under future conditions (G, 2.3% for La). The REE percentage of dissolution increased regularly from La (~2%) to Dy (~5%) and then decreased towards Lu (~3%) (Fig. 5a). It closely mimics the solubility pattern obtained by leaching Saharan aerosols with filtered seawater (Greaves et al., 1994). In this latter study, the mid-REE maximum of percentage of dissolution was attributed to Fe oxyhydroxides (Haley et al., 2004), with Fe oxyhydroxides being the main phase releasing REE, but Fe solubility was not measured. Here, we show that the percentage of REE dissolution exceeds by far the percentage of Fe release, possibly due to the high REE content of Fe oxyhydroxides (Haley et al., 2004).

30

The soluble fraction of Mediterranean aerosols was also evaluated by leaching aerosols collected during the PEACETIME cruise in ultrapure water for 30 min (Fu et al., 2020, this issue). There was a relatively good agreement for Nd between these aerosol leaching experiments (median percentage of Nd dissolution = 6%) and the percentage of Nd dissolution observed in our tank experiments (3%). These low values are also consistent with former estimates based on Saharan aerosol leaching in distilled water (1-3 %; Greaves et al., 1994). In contrast, aerosol leaching during PEACETIME suggested much larger Al and Fe solubilities (around 20%) than those observed during our dust addition experiments. These highest percentages of dissolution reflect mainly the anthropogenic influence in the aerosol samples collected during the cruise that were characterized by mixing between Saharan and polluted air masses (Fu et al., 2020, this issue). Indeed, anthropogenic metals are largely more soluble than metals issued from desert dust (Desboeufs et al., 2005). However, the solubility values obtained in our dust addition experiments for Al and Fe are in agreement with the values found in ultrapure water for the same amended dust (Aghnatios et al., 2014), for other analogs of Saharan dust (Desboeufs et al., 2001; Paris et al., 2011) or for dust collected over Sahara (Paris et al., 2010).



We conclude that the percentages of particulate Al and Fe obtained in our experiments are representative of pure Saharan dust inputs.

### 4.2 Removal of dissolved tracers

5       During the experiments, biological uptake or scavenging onto particulate matter may have biased the estimation of percentage dissolution of insoluble elements. Fe and Al are well known for being incorporated during biological processes: Fe as a micronutrient (e.g. Twinings et al., 2015) and Al by substitution to Si in diatom frustules (Gehlen et al., 2002). Could the lack of increase of dissolved Fe be due to biological uptake? We estimate this uptake with the Chlorophyll $a$ (Chl$a$) increase observed during the course of the experiments ($\sim 0.5$ µg·L$^{-1}$), a C/Chl$a$ ratio of

$\sim 50$ mg C/mg Chl$a$ and a Fe/C ratio of 10-100--70 µmol/mol (Twinings et al., 2015). It yields that the biological activity should have taken up at most 0.25--0.2 nM of Fe, which is an order of magnitude less than the dissolved Fe measured during the course experiments. Therefore, any significant Fe release by dust would not have been masked by biological uptake. We also estimated if the development of diatoms during the experiment can remove a significant fraction of Al from the solution. Using the biogenic silica flux measured in the sediment traps ($\sim 10$-40 mg·m$^{-2}$·d$^{-1}$)

and an Al/Si ratio in diatom frustules of 0.008 (maximal value in Gehlen et al., 2002, to maximize the effect), we calculated that diatoms could have consumed as much as 6-18 nM Si. This represents a small but not completely negligible fraction of the Al released by the dust.

As REE and Th are not known to be involved in biological cycles, their decreasing concentrations during the course

of the experiments suggests that they may be removed by scavenging onto particles due to abiotic processes. We define the percentage of scavenging as follow:

$$f_{scav} = \frac{CONC_{max} - CONC_{min}}{CONC_{max}} \times 100 \qquad (2)$$

Th appeared to be the element most sensitive to scavenging (43-44% at ION and 65-70% at FAST). The REE scavenging was less extensive and decreased from LREE (15-37%) to HREE (1-13%; Fig. 5b). This reduced scavenging of HREE compared to LREE is consistent with the stronger complexation of HREE by carbonate ions that stabilize them in seawater (Tachikawa et al., 1999) and was already observed during equilibrium experiments between seawater and synthetic minerals (Koeppenkastrop and Eric, 1992). In terms of REE fractionation, the net effect of

preferential Mid-Rare Earth Elements (MREE) release (from particles enriched in MREE) and the preferential scavenging of LREE compared to HREE results in a shale-normalized REE pattern with a reduced depletion of LREE compared to HREE, but a flat pattern from MREE to HREE (Fig. ES 2).

      For both Th and LREE, the scavenged fraction is larger at FAST than at ION. At each station, similar scavenging percentages were obtained from D and G experiments. When discussing scavenging, it must be kept in

mind that the large amount of dust initially introduced in the tanks, yielded an unusually high particle content in seawater. Considering that 3.6 g of dust were introduced in 300 L of seawater, that ~50% of the fast-sinking large particles (Bressac et al., 2011) sedimented to the bottom trap and that 15% of carbonate dissolution occurred (see section 3.1, Desboeufs et al., 2014), the average dust concentration remaining in suspension in the tank was $\sim 5000$ µg·L$^{-1}$. This was several orders of magnitude higher than typical particulate matter concentrations in seawater (1-100

µg·L$^{-1}$, Lal, 1977), not impacted by a recent dust deposition event. At these high particulate matter concentrations, it is expected that scavenging of insoluble elements onto suspended dust can occur. Adsorption experiments of a radioactive Ce tracer on deep sea clays showed a decrease of 30 % of the dissolved Ce over a few days (Li et al., 1984), which is grossly comparable to the results presented here. These experiments on deep sea clays were carried in abiotic





conditions, raising the possibility that adsorption observed during the tank experiments were, at least in part, due to abiotic processes on the dust. However, as the dust used during ION and FAST were strictly similar, it is likely that larger and faster scavenging at FAST compared to ION was due to the higher biological activity at FAST compared to ION (Gazeau et al., 2020a, this issue). Among biologically produced molecules, Th has a high affinity for TEP

(Santschi et al., 2006). However, there not a marked difference in TEP content at ION compared to FAST (Gazeau et al., 2020b, this issue**.

### 4.3 Thorium isotopes

When reversible processes affecting Th occur, the $^{230}$Th/$^{232}$Th ratio is a good tracer of the Th fluxes (Roy-Barman et al., 2002). The rationale is that the $^{230}$Th/$^{232}$Th ratio in initial seawater ($^{230}$Th/$^{232}$Th ≈ 15 × 10$^{-6}$, Gdaniec et al., 2018) is higher than the $^{230}$Th/$^{232}$Th ratio in the dust ($^{230}$Th/$^{232}$Th ≈ 3-6 × 10$^{-6}$; Pham et al., 2005; Roy-Barman et al., 2009). Hence, as dust releases dissolved Th in the tanks, the seawater $^{230}$Th/$^{232}$Th ratio decreases. Conversely, when dissolved Th is scavenged on the particles, both isotopes behave similarly and the isotopic ratio of the seawater inside

the tank remains constant. Hence the $^{230}$Th/$^{232}$Th ratio of the seawater keeps track of the Th released from dust even if some readsorption occurs. As there is far more Th in the dust than in the initial seawater, the $^{230}$Th/$^{232}$Th ratio of the particulate matter remains virtually constant even if seawater-derived Th sorbs on the particles.

These changes in concentration and isotopic ratios are best illustrated by plotting the $^{230}$Th/$^{232}$Th ratio as a

function of 1/$^{232}$Th (Fig. 6). On this diagram, the theoretical evolution of the filtered seawater samples with time should be: (1) for simple dissolution, filtered seawater samples lie on a straight line between labile marine particles and seawater blanks; (2) if readsorption occurs, the filtered seawater samples will be shifted horizontally toward the right; (3) for rapid reversible equilibrium between seawater and particles, filtered seawater samples should lie on a straight vertical line (Arraes-Mescoff et al., 2001). For the ION experiments, C samples and the samples of the D and G

treatments at t = 1 h plot on an oblique straight line, suggesting that the initial increase in seawater Th concentration results from the simple dissolution of marine particles. On this diagram, the intercept at 1/$^{232}$Th = 0 represents the $^{230}$Th/$^{232}$Th of the dissolving particulate matter. It appears that this ratio is about $(8.5 \pm 0.8) \times 10^{-6}$ mol/mol, significantly above the ratio measured on the trapped particles ($\sim(6 \pm 0.5) \times 10^{-6}$) or estimated for Saharan dust (Pham et al., 2005, Roy-Barman et al., 2009). It suggests that there might be a preferential release of $^{230}$Th compared to $^{232}$Th (Bourne et

al., 2012, Bosia et al., 2018, Marchandise et al., 2014).
Samples from the D and G treatments (t = 1 h to 72 h) plot on a horizontal line (with little change of the $^{230}$Th/$^{232}$Th ratio) confirming that simple reabsorption occurs after the initial dissolution (with little or no release of particulate Th after the initial dissolution observed at t = 1 h).

A simple mass balance gives the fraction of dissolved $^{232}$Th in seawater coming from the dissolution of particulate Th (Roy-Barman et al., 2002):

$$f_{litho} = \frac{\left(\frac{230Th}{232Th}\right)_{D\ or\ G} - \left(\frac{230Th}{232Th}\right)_{C}}{\left(\frac{230Th}{232Th}\right)_{litho} - \left(\frac{230Th}{232Th}\right)_{C}} \qquad (3)$$

Knowing $f_{litho}$, we can determine $f_{dissol\_isot}$, the dissolution fraction based on the isotopic data:



$$f_{diss\_isot} = \frac{CONC_{init}\left(\frac{f_{litho}}{1 - f_{litho}}\right)}{CONC_{dust} m/V} \tag{4}$$

This estimate is independent on the concentration data that may be biased by readsorption. We evaluate $f_{dissol\_isot}$ based on average ratios of the C series for original seawater and the average ratio of the last samples of the D and G treatments

to integrate dissolution over the course of the whole experiment (Tab. ES3). For the particulate ratio, we tentatively used a ratio of $8.5 \times 10^{-6}$ mol/mol (value best defined by the y-axis value for $1/^{232}$Th = 0 of the dissolution and scavenging trends at ION, Fig. 5).

The resulting average $f_{dissol\_isot}$ are below 3% for FAST D, FAST G, ION D and ION G, confirming the low solubility of Th (Tab. 1). While we recognize that for FAST, the large data scattering results in large uncertainties on the

interpretation of the results, all the results obtained during PEACETIME argue for a low solubility of Th.

**4.4 implication for dust deposition estimation**

      The $^{230}$Th/$^{232}$Th ratio in the surface ocean is proposed as a tracer to monitor the dust inputs at the ocean surface (Hsieh et al., 2012). The rational is that, neglecting lateral transport, $^{232}$Th is provided by dust dissolution whereas

$^{230}$Th comes mostly from the *in situ* decay of $^{234}$U and can be used as a chronometer for $^{232}$Th dissolution. If the solubility of $^{232}$Th from dust is known, it is possible to calculate the dust flux required to account for the $^{230}$Th/$^{232}$Th ratio of the surface waters. Until now, the $^{232}$Th solubility from dust was poorly constrained. By adjusting the percentage of lithogenic $^{232}$Th dissolution to match the $^{230}$Th/$^{232}$Th ratio in the surface water of the Atlantic Ocean and using dust fluxes from a global dust deposition model, it was proposed that the fraction of lithogenic $^{232}$Th dissolution

grossly range between 1 and 5% in high dust flux areas such as the East equatorial Atlantic and up to 10-16 % in areas of low dust deposition such as the South Atlantic (Hsieh et al., 2012). Estimated $^{232}$Th percentage of dissolution at the Bermuda Atlantic Time-series Study (BATS) ranged from ~1 to 68% and increased with the assumed aerosol dissolution depth range (Hayes et al., 2017). Alternatively, the soluble fraction of $^{232}$Th in atmospheric dust was estimated by leaching particles with deionized water or dilute acetic acid, but the 2 types of leaching provided no

consensus results (Anderson et al., 2017).

The present work indicates a low $^{232}$Th solubility (percentage of dissolution ~1%) for the material used for the tank experiments. It is quantitatively consistent with the low solubility (percentage of dissolution~3-5%) of lithogenic Th derived from a gross western Mediterranean budget of Th isotopes in which Th inputs are dominated by ocean margins (Roy-Barman et al., 2002). Keeping in mind the limitation of our study (limited time duration, very high particle

concentration promoting re-adsorption), these results argue for a low $^{232}$Th solubility. It suggests that the high Th solubility derived by balancing the lithogenic Th input by dust with the scavenging on settling particles may be strongly biased by advective inputs (Hayes et al., 2017). Hence, deducing dust inputs from the $^{230}$Th/$^{232}$Th ratio of surface waters (Hsieh et al., 2012) cannot be done with a simplified 1-dimensional view and requires to take lateral transport into account.






## 5 Conclusion

The PEACETIME tank experiments have allowed to quantify the particulate-dissolved exchanges of Al, Fe, REE, Pa and Th following Saharan dust addition to surface seawater in three basins of the Mediterranean Sea under present and future climate conditions. In particular, we report here the first estimates of thorium and protactinium. We highlight differences of percentages and kinetics of dissolution as well as scavenging among the lithogenic tracers: under the experimental conditions, Fe dissolution was much more buffered than the dissolution of Th, REE or Al. As a consequence, assuming a congruent dissolution of lithogenic tracers to evaluate Fe fluxes is probably generally not appropriate. Using relative solubility, might be also biased by the different dissolution and scavenging kinetics characterizing each tracer. Quite unexpectedly, comparison of present and future conditions indicates that changes in temperature (+3 °C) and/or pH (-0.3 pH unit) influence the release of thorium and REE in seawater, leading to a lower Th release and a higher light REE release under increased greenhouse conditions. Using Th isotopes, we show that Th was released within the first hour of the experiment and that no subsequent Th release occurs during the following days. This observation, associated to the low percentage (1%) of Th dissolution from dust, puts strong constraints on the use of Th isotopes as a tracer of dust inputs in surface waters and highlights the importance of advection as a source of $^{232}$Th is the open ocean.

The implications of these experiments are not limited to constrain aeolian inputs to the surface ocean. They also contribute to a better understanding of the strong contrast in vertical profiles and zonal distribution of insoluble elements in the Mediterranean Sea. In this region, dissolved Al increases from surface to deep waters and also from the deep western basin to the deep eastern basin (Rolison et al., 2015), whereas dissolved Fe and $^{232}$Th profiles often present surface concentration maxima and no systematic concentration gradient between the deep western and deep eastern basins (Gerringa et al., 2017; Gdaniec et al., 2018). While the Al and Th percentages of dissolution from dust were comparable during the tank experiments, kinetics were different: Th strongly scavenged after the initial release, whereas Al kept dissolving all time long. It highlights the highly particle-reactive character of Th compared to Al. Hence $^{232}$Th, cannot accumulate along the Mediterranean deep circulation and does not exhibit a zonal gradient like Al does.

**Data availability**

Underlying research data are being used by researcher participants of the "Peacetime" campaign to prepare other manuscripts and therefore data are not publicly accessible at the time of publication. Data will be accessible (http://www.obs-vlfr.fr/proof/php/PEACETIME/peacetime.php, last access: 22 June 2020) once the special issue is completed (all papers should be published by fall 2020). The policy of the database is detailed here http://www. obs-vlfr.fr/proof/dataconvention.php (last access: 22 June 2020).

**Author contribution**

CG, FG and KD conceived the PEACETIME program and the tank experiments. MB analyzed dissolved Fe. TW analyzed dissolved Al. NL analyzed trapped material, MRB, LF and ED analyzed Th, Pa and REE. MRB prepared the manuscript with contributions from all co-authors.





**Competing interests**

The authors declare that they have no conflict of interest, no competing financial interests.

**Acknowledgements**

This study is a contribution to the PEACETIME project (http://peacetime-project.org), a joint initiative of the
5    MERMEX and ChArMEx components supported by CNRS-INSU, IFREMER, CEA, and Météo-France as part of the
MISTRALS program coordinated by INSU (PEACETIME cruise https://doi: 10.17600/17000300). All data have been
acquired during the PEACETIME oceanographic expedition on board R/V Pourquoi Pas? in May-June 2017.
PEACETIME was endorsed as a process study by GEOTRACES. M.B was funded from the European Union Seventh
Framework Program ([FP7/2007-2013]) under grant agreement no. [PIOF-GA-2012-626734] (IRON-IC project)). We
10   thank the captain and the crew of the RV Pourquoi Pas? for their professionalism and their work at sea.



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



Table 1: Maximum percentage of dissolution and percentage of scavenging

| | Fe | Al | La | Ce | Pr | Nd | Sm | Eu | Gd | Tb | Dy | Ho | Er | Tm | Yb | Lu | Th_conc | Th_isot* | Pa |
|---|---|---|---|---|---|---|---|---|---|---|---|---|---|---|---|---|---|---|---|
| Percentage of dissolution (%) | | | | | | | | | | | | | | | | | | | |
| TYR_D | 0.012 | 0.56 | | | | | | | | | | | | | | | | | |
| TYR_G | 0.02 | 0.82 | | | | | | | | | | | | | | | | | |
| ION_D | 0.07 | 0.91 | 1.9 | 2.2 | 2.8 | 3.6 | 4.0 | 4.5 | 4.4 | 4.9 | 5.3 | 5.5 | 5.1 | 4.4 | 4.1 | 4.1 | 1.0 | 0.69 (0.64-0.73) | |
| ION_G | 0.03 | 0.96 | 2.2 | 2.5 | 3.0 | 3.7 | 4.0 | 4.3 | 4.5 | 4.8 | 5.1 | 5.4 | 4.9 | 4.4 | 3.8 | 3.7 | 0.7 | 1.1 (1.0-1.2) | |
| FAST_D | 0.10 | 1.09 | 1.8 | 2.3 | 3.0 | 3.7 | 4.2 | 4.7 | 5.7 | 5.4 | 5.6 | 5.6 | 5.3 | 4.5 | 4.1 | 4.1 | 1.2 | 0.1 (0-8) | < 6% |
| FAST_G | 0.04 | 1.13 | 2.3 | 2.5 | 3.1 | 3.8 | 4.1 | 4.3 | 4.6 | 5.1 | 5.5 | 5.9 | 5.3 | 4.4 | 4.1 | 3.6 | 0.9 | 2.4 (1-7) | |
| | | | | | | | | | | | | | | | | | | | |
| Percentage of scavenging (%) | | | | | | | | | | | | | | | | | | | |
| ION_D 72h | | | 14 | 11 | 20 | 21 | 22 | 20 | 14 | 12 | 7.3 | 4.0 | 1.4 | -1.2 | -0.3 | 0.1 | 43 | | |
| ION_G 72h | | | 9 | 8 | 19 | 20 | 20 | 18 | 15 | 12 | 8.7 | 4.6 | 3.2 | 2.9 | 0.9 | 0.7 | 44 | | |
| FAST_D 72h | | | 25 | 27 | 31 | 29 | 29 | 27 | 23 | 21 | 15 | 7.8 | 4.7 | 1.6 | 4.2 | 3.0 | 60 | | |
| FAST_D 96h | | | 18 | 36 | 37 | 36 | 34 | 35 | 29 | 27 | 18 | 10 | 6.8 | 4.6 | 12.8 | 3.6 | 65 | | |
| FAST_G 96h | | | 18 | 23 | 21 | 23 | 25 | 20 | 17 | 14 | 9 | 5 | 0.5 | -1 | -1 | -7 | 70 | | |

The percentage of dissolution is calculated according to equation 1, except for $Th_{isot}$ which is based on equation 4. The percentage of scavenging is calculated according to equation 2. For 232Th, we did not take samples D1-12h and D1-24h, because they were considered contaminated (section 3.4). Taking theses samples into account would not change qualitatively the main conclusions of the study on thorium.





**Figure caption:**

**Figure 1**: concentrations of total dissolved Fe and Al during the dust addition experiments

5    **Figure 2**: Concentrations of dissolved REE during the tank experiments.  a) ION station. b) FAST station.

**Figure 3**: Dissolved $^{232}$Th during the ION and FAST experiments. Note the scale brake to show the 3 outliers (contaminated samples).

10   **Figure 4**: Dissolved Pa during the FAST experiments. Error bars correspond to the analytical uncertainties.

**Figure 5:** Average maximum percentage of dissolution and percentage of scavenging of REE.  (a) Percentage of dissolution as defined by equ. 1. Percentage of scavenging as defined by equ. 2. For purpose of comparison, percentage of scavenging was calculated only with data at t=72h for both ION and FAST.

**Figure 6**: $^{230}$Th/$^{232}$Th versus 1/$^{232}$Th mixing diagram for ION (left) and FAST (right) experiments. Seawater data pooled by time since dust addition: blue dots: t = 0 h (no dust addition yet); red dots: t = 1-6 h; orange dots: t = 24-48 h; yellow dots: t = 72-96 h. Yellow dots between brackets fall above the scavenging line for an unknown reason. Green dots: particles in the sediment traps. Red arrow: preferential release of $^{230}$Th.



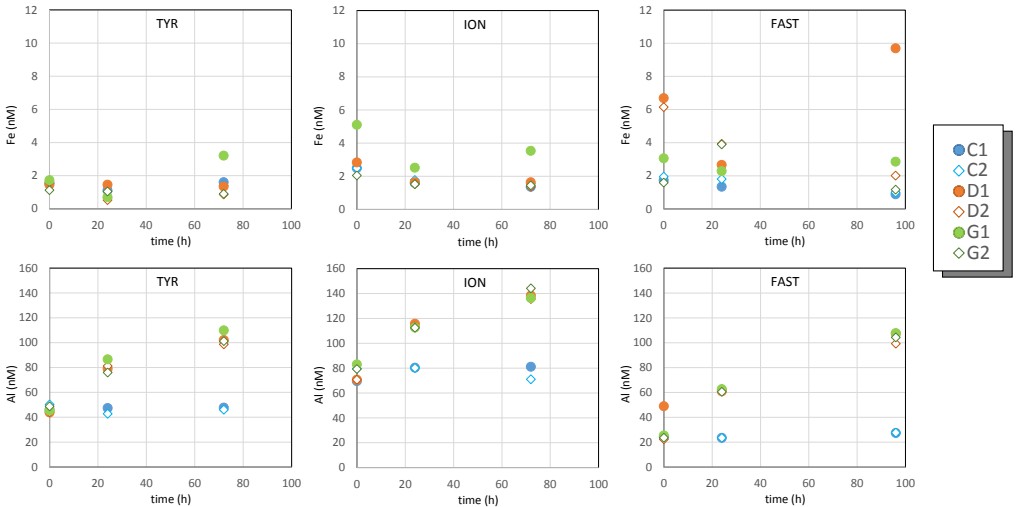

10    Fig 1



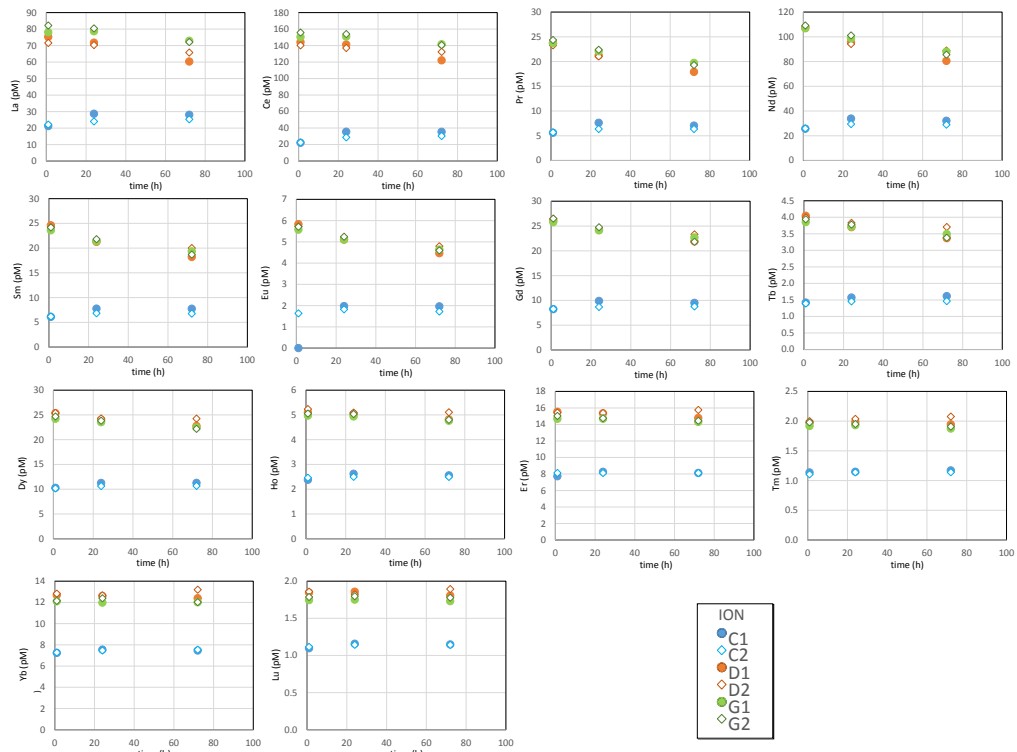

Fig 2a.



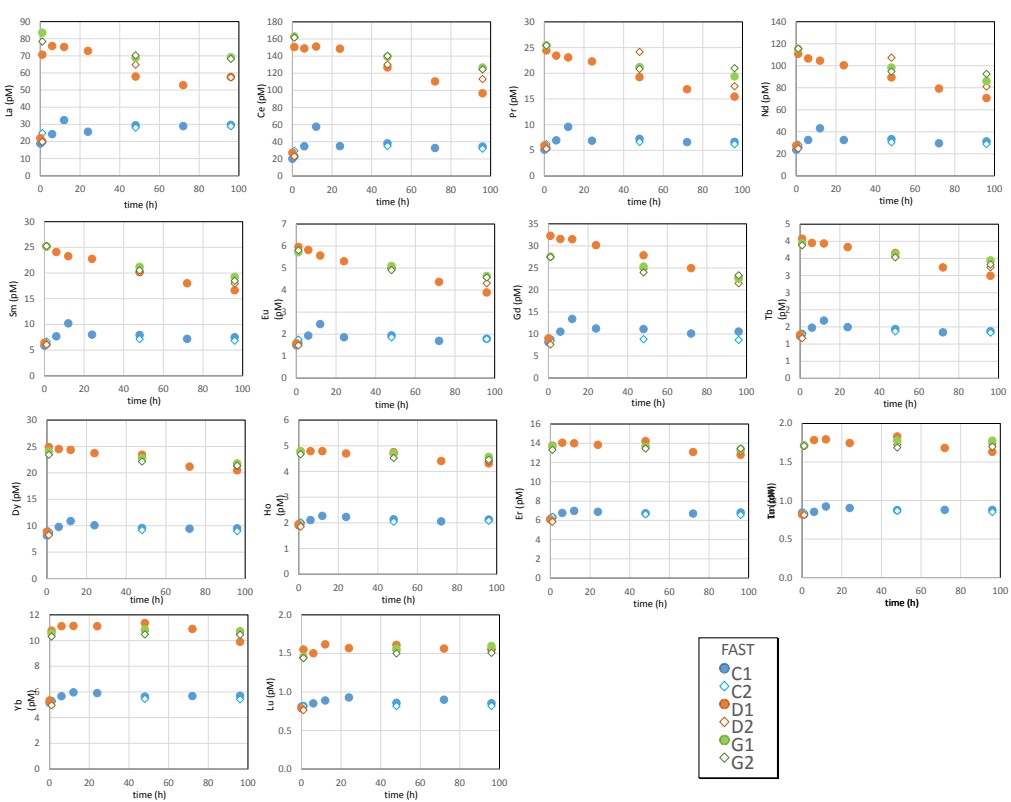

15    Fig 2b.





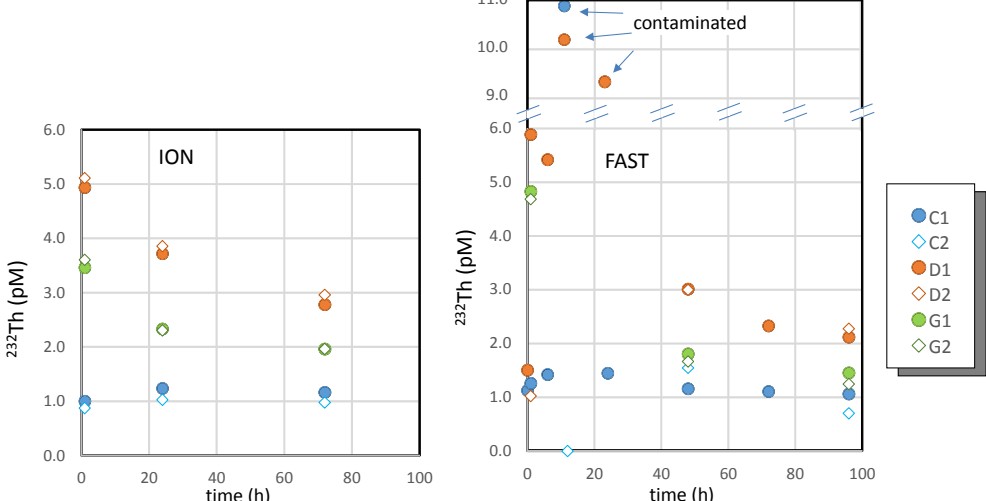

Fig. 3



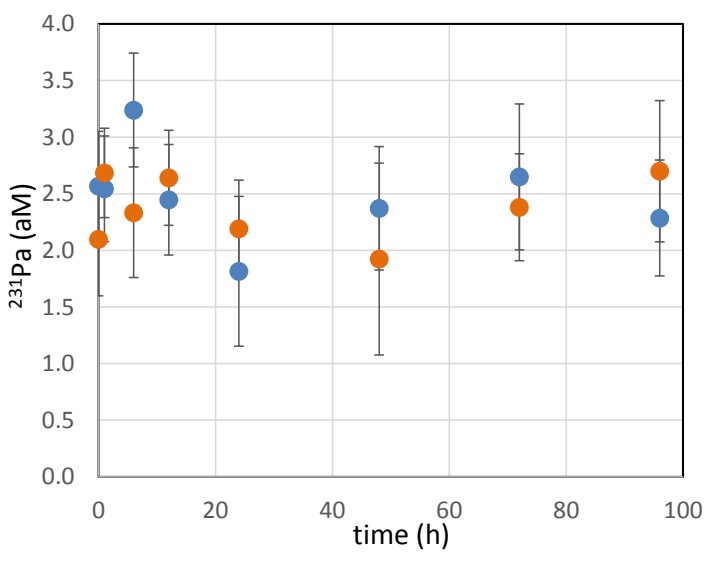

15    Fig. 4:





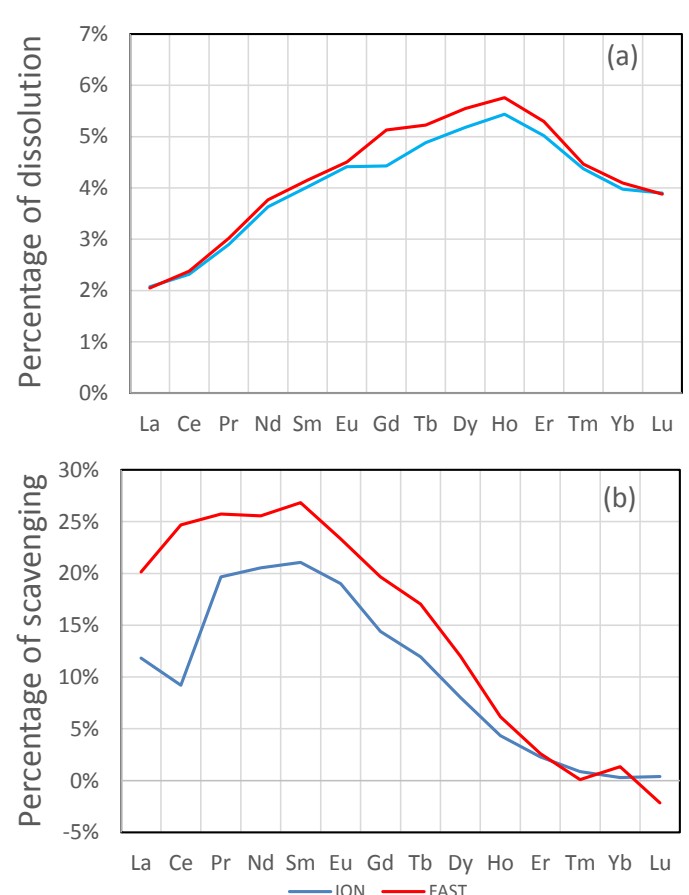

Fig. 5.


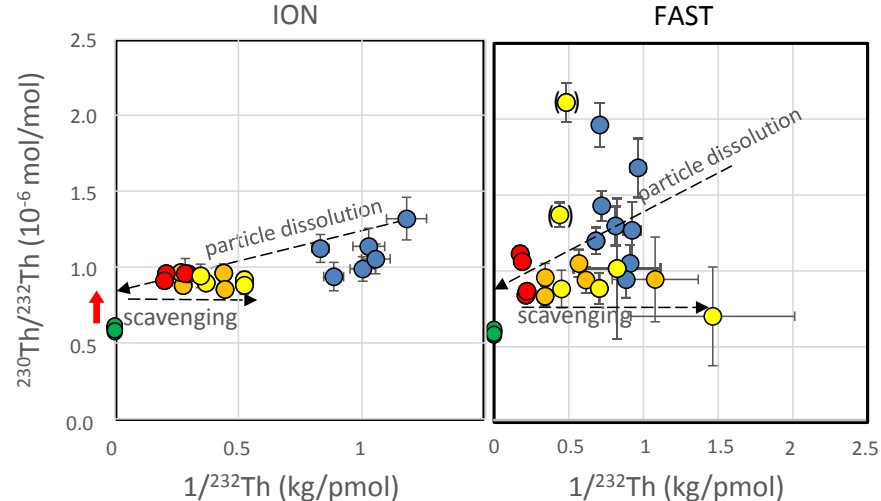

Fig. 6