# Peer review of "Contrasted release of insoluble elements (Fe, Al, REE, Th, Pa) after dust deposition in seawater: a tank experiment approach"

_Biogeosciences, 2020_

## Referee Comment (RC1) · Anonymous Referee #1 · 6 Aug 2020

This is a creative and timely study on the solubility of dust-derived elements in the ocean. The authors find a range of behavior among different elements which will certainly help oceanographers interpret biogeochemical cycling of these elements. This is a high quality and careful study that should be published in Biogeosciences. I have a few substantive comments for the authors to address before final publication and several technical comments.

Substantive comments: Line 35, page 3. More needs to be said about how it was determined what is a "realistic" dust input. The 10 g/m2 is realistic over what timescale? If considered over a 3 day time period, this would correspond to a flux of ∼1200 g/m2/yr,

which I believe is on the high end of model-based estimates for the Mediterranean. If the flux is considered only over an initial time period after dust deposition the effective flux would be even higher. Some of this information is brought up in the discussion, but it should be discussed in the introduction so we understand the experimental design. Could you speculate on how the results might be different in a scenario where a smaller amount of dust was added at regular intervals over a longer time period? It might also be worth mentioning that dust particles fall into low pH, high $CO_2$ and cold water in the thermocline which could affect element solubility.

L2, page 9. Fe ligands are almost always found in excess of observed Fe concentrations (e.g, Buck et al., 2015; https://doi.org/10.1016/j.dsr2.2014.11.016). This suggests Fe is not at a solubility limit if the availability of ligand is the effective solubility limit for seawater. Have Fe ligands been determined in the tanks or in the region?

Page 9. Is it possible to assess an uncertainty in the derived solubility fractions?

Sections 4.4 and 5. There is an interesting juxtaposition of ideas here that should be clarified. First, there is a suggestion that previous high Th solubility fractions could be the result of lateral advective inputs. However, the observations of scavenging within the tank experiments as well as the conclusion that 232Th cannot accumulate along the Mediterranean deep circulation, would suggest relatively little lateral transport of 232Th. There may be certain distance scales the authors have in mind to draw the distinction between these two cases, and if possible, they should be stated more explicitly.

Detailed comments:

Line 14, page 3. Technically 230Th is produced from the decay of 234U.

Line 18, page 3. Pa is defined in the first paragraph so the full name and extra open parenthesis can be removed here.

Lines 18-19, page 3. The meaning of the parenthetical is not clear " (absolute or at least relative) ". I think this could simply be deleted.

Line 28, page 3. Tells us which season is characterized by strong stratification (summer?)

L15, page 4. The concentration given is unclear. Is the final concentration of formaldehyde a weight percentage, volume percentage or mole percentage?

L16, page 5. Missing superscript 232

L23, page 6. Extra E, HREEE

L33, page 8. The equation involving "x m/V" needs to be explained in words. It is not clear what is being calculated here.

L10, page 9. DIP acronym not defined.

L10, page 10. The number/range given is not clear "10-100—70 umol/mol". Is this a range of ratios or a range of numerator and constant denominator?

─────────────────────────────

---

## Referee Comment (RC2) · Frank Pavia (Referee) · 14 Aug 2020

General comments:

The manuscript by Roy-Barman et al. will be an excellent contribution to Biogeosciences, and to chemical oceanography in general. The work constitutes a novel way to estimate the mobility of mostly-insoluble elements during dust dissolution, and the findings are highly relevant to a bourgeoning literature on using Th isotopes to quantify dust dissolution and trace element inputs. The methods, data quality, and interpretations are generally sound. Before publication, there are a several points that need to be addressed, and I encourage the authors to reconsider alternate possibilities for

their recommendations for the use of dissolved Th isotopes to quantify dust fluxes (see general comments below).

Specific comments:

1) I had a hard time following the REE section of the results (section 3.3.). I think this is because of the way Figure 2 is plotted. The sharp increase in REE at the start of the FAST experiment is difficult to see in Figure 2b because the t=0 and t=1 points look like they are at the same place on the x-axis. Maybe it would be good to label the t=1 point as being immediately after dust addition (if this is the case?). It should also be clarified that the rapid increase in REE is only seen at FAST, and not at ION (unless I am having a hard time seeing this at ION because of the same plotting issue).

This same point holds true for 232Th and Figure 3. Also, page 7 line 2 suggests that 230Th/232Th ratios as a function of time are plotted in Figure 3, but I do not see them there – I would quite like to see these ratios plotted with sampling time!

2) I'm curious from an analytical perspective, though not necessarily for the paper, where they think their high 231Pa blanks might come from, since there is so little 231Pa in nature. That said, the precision the authors achieved on such small quantities of 231Pa, in such small samples, is remarkable.

3) I am curious about the inferred dissolution of detrital calcium carbonate in the dust mentioned in 3.6. even though this isn't the main point of the paper. This process must be occurring above the calcite saturation horizon – what is driving this dissolution?

4) I struggle with the arguments in the first paragraph of section 4 (page 7, particularly lines 34-41). What time interval are these waters seeded over? The dust input units given of 10g dust/m2 are not flux units without a time component, so it is difficult to assess whether this is a feasible dust flux compared to the highest dust pulses or mean annual dust fluxes. Because the dust corresponding to the average areal input was added in such a short time period, I think the sentence in lines 37-38 "Hence, the

PEACETIME experiments. . .” must be taken out.

5) The difference in dissolution kinetics between Th/REE and Al is intriguing, and I would like to have seen the authors speculate on what drives it. Are different mineral phases releasing Th/REE than Al? A back of the envelope calculation, 3.6g dust added, 15% of it presumed to be CaCO3 that dissolves, 0.5ppm hydrogenous 232Th in lacustrine carbonates (Lin et al., 1996) and in pedogenic carbonates (Ludwig and Paces, 2002) gives 1160 pmol of 232Th that could be released into the tank from detrital CaCO3 dissolution. Dividing this by the 300L tank volume gives a $\Delta$[232Th] of $\sim$3.9pM – very close to the 3-5 pM initial rapid increase in 232Th observed at both ION and FAST. This calculation should be done by the authors for REE as well. If this is the case, then Th dissolution in dust may be highly dependent on the detrital carbonate content – which would be a really interesting result. Regardless, it is important to explicitly note that Th and REE release from dust is clearly controlled by multiple phases – a fast dissolving one and a slow-dissolving one – and that this is different from Fe and Al release.

6) The relative fractional solubility of metals is used in calculations of dissolved metal flux from dust derived from thorium isotopes (Hayes et al., 2018; Pavia et al., 2020). It would be fantastic to calculate the release ratios of Fe, Al, and REE to Th (along with their uncertainties) and provide these in addition to Table 1. These would be an excellent addition to the aerosol-based solubility estimates currently used, though it would also be important to note that using these relative solubilities in Th-based Fe flux estimates could be biased if the differences in metal release kinetics between Fe and Th release are sufficiently large to decouple the release depths of Fe and Th as fine lithogenic particles aggregate and sink from surface waters.

7) There is a mismatch by a factor of 10 between the units on y-axis of Figure 6 and those cited in the text in section 4.3.

8) The points regarding differences in solubility estimates of Th between previous results and this study made in 4.4. are very important. Th solubility is one of the largest sources of uncertainty in Th-based dust flux estimates (Pavia et al., 2020), and different methods give different results. However, authors' points about lateral 232Th inputs are not justified by the data presented in this paper. The solubility of 232Th does not alter the removal residence time of Th in the upper water column. The 230Th-based removal residence times in the upper 250m of the South Pacific Gyre (depths over which Th-based dust flux estimates are integrated) are 1-2 years (Pavia et al., 2020). Note that the South Pacific Gyre is highly oligotrophic, similar to the Mediterranean – so 1-2 years is likely on the longer end of Th residence times for the upper 250m in the ocean. The diffusive lengthscale for lateral diffusion of Th in the upper water column, given by $\Delta x = \sqrt{(2K\_H\,\Delta t)}$ (Roy-Barman, 2009), for a residence time of 2 years and a lateral eddy diffusivity of 10ˆ3 m2/s, is 355km. So, regardless of the 232Th solubility, lateral lithogenic inputs from the continents should not affect 232Th distributions in the open ocean beyond this distance. Indeed, the lack of lateral inputs affecting the deep Mediterranean, with longer transport timescales from boundary lithogenic sources than in the upper water column in the open ocean, is noted in the conclusion. More work is needed to determine the role of vertical and lateral exchange on 230Th and 232Th budgets, but the conclusions of this study do not justify the conclusiveness of the statements regarding lateral Th fluxes (e.g. "cannot be done", page 12 line 33; "strongly biased by advective inputs" page 12 lines 31-32). These statements should be tempered accordingly. Similarly, the statement on page 13 lines 13-15 should be adjusted or removed, as the current study does not provide any insights on the importance of advection as a source of 232Th to the open ocean.

References

Hayes, C. T., Anderson, R. F., Cheng, H., Conway, T. M., Edwards, R. L., Fleisher, M. Q., Ho, P., Huang, K.-F., John, S. G., Landing, W. M., Little, S. H., Lu, Y., Morton, P. L., Moran, S. B., Robinson, L. F., Shelley, R. U., Shiller, A. M. and Zheng, X.-Y.: Replacement Times of a Spectrum of Elements in the North Atlantic Based on Thorium Supply, Global Biogeochemical Cycles, 32(9), 1294–1311, doi:10.1029/2017GB005839, 2018. Lin, J. C., Broecker, W. S., Anderson, R. F., Hemming, S., Rubenstone, J. L. and Bonani, G.: New 230Th/U and 14C ages from Lake Lahontan carbonates, Nevada, USA, and a discussion of the origin of initial thorium, Geochimica et Cosmochimica Acta, 60(15), 2817–2832, doi:10.1016/0016-7037(96)00136-6, 1996. Ludwig, K. R. and Paces, J. B.: Uranium-series dating of pedogenic silica and carbonate, Crater Flat, Nevada, Geochimica et Cosmochimica Acta, 66(3), 487–506, doi:10.1016/S0016-7037(01)00786-4, 2002. Pavia, F. J., Anderson, R. F., Winckler, G. and Fleisher, M. Q.: Atmospheric Dust Inputs, Iron Cycling, and Biogeochemical Connections in the South Pacific Ocean from Thorium Isotopes, Global Biogeochemical Cycles, doi:10.1029/2020GB006562, 2020. Roy-Barman, M.: Modelling the effect of boundary scavenging on Thorium and Protactinium profiles in the ocean, Biogeosciences Discussions, 6(4), 7853–7896, doi:10.5194/bgd-6-7853-2009, 2009.

---

## Author Comment (AC1) · 15 Oct 2020

**Answer to anonymous Referee #1**

*We thank the anonymous reviewer for her/his comments and suggestions on our manuscript. We agree with most comments and modified/updated the manuscript accordingly. Below is a point-by-point reply, [our answers appear in italics].*

This is a creative and timely study on the solubility of dust-derived elements in the ocean. The authors find a range of behavior among different elements which will certainly help oceanographers interpret biogeochemical cycling of these elements. This is a high quality and careful study that should be published in Biogeosciences. I have a few substantive comments for the authors to address before final publication and several technical comments.

**Referee question:**
Substantive comments: Line 35, page 3. More needs to be said about how it was determined what is a "realistic" dust input. The 10 g/m2 is realistic over what timescale? If considered over a 3 day time period, this would correspond to a flux of _1200 g/m2/yr, which I believe is on the high end of model-based estimates for the Mediterranean. If the flux is considered only over an initial time period after dust deposition the effective flux would be even higher. Some of this information is brought up in the discussion, but it should be discussed in the introduction so we understand the experimental design.

*Reply: It is now indicated in the introduction and sections 2.1. & 4. that dust enrichment occurred at the beginning of the experiment and that it corresponds to the simulation of a strong Saharan dust deposition event in the Mediterranean Sea, as illustrated by Ternon et al., (2010).*
*Ternon, E., Guieu, C., Loÿe-Pilot, M. D., Leblond, N., Bosc, E., Gasser, B., Miquel, J. C., Martin, J.: The impact of Saharan dust on the particulate export in the water column of the North Western Mediterranean Sea, Biogeosci., 7, 809–826, https://doi.org/10.5194/bg-7-809-2010, 2010.*

**Referee #1:** Could you speculate on how the results might be different in a scenario where a smaller amount of dust was added at regular intervals over a longer time period?

*Reply: Done. We now indicate in section 4.2 that very high dust content and adsorption rates were reached because all the dust was deposited instantaneously at the beginning of the experiment. Deposition of the same amount of dust over longer periods (weeks, months) as it occurs in les dusty environment than the Mediterranean Sea, would certainly result in less readsorption (but likely similar percentages of dissolution).*

**Referee #1:** It might also be worth mentioning that dust particles fall into low pH, high CO2 and cold water in the thermocline which could affect element solubility.

*Reply: It is complicated to extrapolate these surface water results to the thermocline, because it is unclear if the change in dissolution fraction is a direct effect of temperature or $CO_2$ change or an indirect effect through the biological activity. Therefore it is not possible to extrapolate to the thermocline where T, $CO_2$ and biological activity will not covariate like during the present experiment.*

**Referee #1:** L2, page 9. Fe ligands are almost always found in excess of observed Fe concentrations (e.g, Buck et al., 2015; https://doi.org/10.1016/j.dsr2.2014.11.016). This suggests Fe is not at a solubility limit if the availability of ligand is the effective solubility limit for seawater. Have Fe ligands been determined in the tanks or in the region?

*Reply: We agree with this comment of the reviewer. The effective solubility of iron is controlled by the availability of ligand in seawater and the role of Fe ligands in maintaining high concentration of DFe in seawater has been emphasized in many studies. Iron binding ligands have not been determined in the mesocosm. Only a limited amount of measurements have been done during the deployment of the RESPIRE traps during the cruise (Bressac et al 2020, Whitby et al 2020) indicating ligand excess in the subsurface. However, the limited existing studies on iron binding ligand concentration in the Med. Sea confirms the ligand excess in the surface Med. Sea. (e.g. Gerringa et al. 2017, Wuttig et al. 2013, Wagener et al 2008, Van den Berg, 1995) We mention the paper by Wagener et al. 2008 that demonstrates seasonal variation of organic speciation for iron and reports excess ligand concentration values for July of the same order of magnitude compared to a previous study by Van den Berg, 1995 (total iron binding ligands were in the range for both studies: 4.2-5.85 nM-eqFe).*

*Bressac, M., Guieu, C., Ellwood, M. J., Tagliabue, A., Wagener, T., Laurenceau-Cornec, E. C., et al. (2019). Resupply of mesopelagic dissolved iron controlled by particulate iron composition. Nature Geoscience, 12(12), 995–1000. https://doi.org/10.1038/s41561-019-0476-6*

*Gerringa, L. J. A., Slagter, H. A., Bown, J., van Haren, H., Laan, P., de Baar, H. J. W., & Rijkenberg, M. J. A. (2017). Dissolved Fe and Fe-binding organic ligands in the Mediterranean Sea–GEOTRACES G04. Marine Chemistry, 194, 100–113. https://doi.org/10.1016/j.marchem.2017.05.012*

*van den Berg, C. M. G. (1995). Evidence for the organic complexation of iron in seawater. Marine Chemistry, 50(1–4), 139–157.*

*Whitby, H.; Bressac, M.; Sarthou, G.; Ellwood, M. J.; Guieu, C.; Boyd, P. W. Contribution of Electroactive Humic Substances to the Iron-Binding Ligands Released During Microbial Remineralization of Sinking Particles. Geophysical Research Letters 2020, 47 (7), e2019GL086685. https://doi.org/10.1029/2019GL086685.*

*Wagener, T., Pulido-Villena, E., & Guieu, C. (2008). Dust iron dissolution in seawater: Results from a one-year time-series in the Mediterranean Sea. Geophysical Research Letters, 35(16).*

*Wuttig, K., Wagener, T., Bressac, M., Dammshäuser, A., Streu, P., Guieu, C., & Croot, P. (2013). Impacts of dust deposition on dissolved trace metal concentrations (Mn, Al and Fe) during a mesocosm experiment. Biogeosciences, 4, 2583–2600.*

**Referee #1:** Page 9. Is it possible to assess an uncertainty in the derived solubility fractions?

*Reply: The propagated uncertainties on the solubility factions are now provided in table 1.*

**Referee #1:** Sections 4.4 and 5. There is an interesting juxtaposition of ideas here that should be clarified. First, there is a suggestion that previous high Th solubility fractions could be the result of lateral advective inputs. However, the observations of scavenging within the tank experiments as well as the conclusion that $^{232}$Th cannot accumulate along the Mediterranean deep circulation, would suggest relatively little lateral transport of $^{232}$Th. There may be certain

distance scales the authors have in mind to draw the distinction between these two cases, and if possible, they should be stated more explicitly.

*Reply: Scavenging within the tank is enhanced by the very high particle concentration. Although the dust quantity deposited in the tanks was realistic for the Mediterranean Sea (observed in situ, see for ex Ternon et al., 2010), it is not relevant for less "dusty" areas of the ocean and the high dust concentration observed in the amended tanks cannot be extended to all ocean surface waters. Therefore, the short Th residence time implied by the fast Th scavenging in the amended tanks is relevant for the Mediterranean Sea after strong dust deposition, but certainly not for most open ocean areas where the Th residence time is longer (as measured by $^{234}$Th in surface waters for example or, as mentioned by Frank Pavia in his review "The 230Th-based removal residence times in the upper 250m of the South Pacific Gyre (depths over which Th-based dust flux estimates are integrated) are 1-2 years (Pavia et al., 2020)."). Hence the good scale necessary to avoid lateral transport, as developed in the answer to the review of Frank Pavia, is , at best, the gyre scale.*
*We agree that $^{232}$Th does not accumulate along the deep Med sea circulation, but we stress that in the Med sea, $^{232}$Th input and scavenging potentially occur all along this path through exchanges with the nearby margins.*

Detailed comments:
**Referee #1:** Line 14, page 3. Technically 230Th is produced from the decay of 234U.

*Reply: This is changed*

**Referee #1:** Line 18, page 3. Pa is defined in the first paragraph so the full name and extra open parenthesis can be removed here.

*Reply: This is done*

**Referee #1:** Lines 18-19, page 3. The meaning of the parenthetical is not clear " (absolute or at least relative) ". I think this could simply be deleted.

*Reply: This is deleted*

**Referee #1:** Line 28, page 3. Tells us which season is characterized by strong stratification (summer?)

*Reply:* It is now indicated that the cruise occurred during the late spring.

**Referee #1:** L15, page 4. The concentration given is unclear. Is the final concentration of formaldehyde a weight percentage, volume percentage or mole percentage?

*Reply: It is now indicated in the text that volume percentage are considered*

**Referee #1:** L16, page 5. Missing superscript 232

*Reply: This is corrected*

**Referee #1:** L23, page 6. Extra E, HREEE

*Reply: This is corrected*

**Referee #1:** L33, page 8. The equation involving "x m/V" needs to be explained in words. It is not clear what is being calculated here.

*Reply: It is now explained that m represents the mass of dust added to the tank and V represents the volume of seawater in the tank.*

**Referee #1:** L10, page 9. DIP acronym not defined.

*Reply: Dissolved Inorganic Phosphorus is now defined*

**Referee #1: Referee #1:** L10, page 10. The number/range given is not clear "10-100ă˘Ă˘T70 umol/mol". Is this a range of ratios or a range of numerator and constant denominator?

*Reply: Clarified by indicating that the Fe/C ratio ranges from 10 μmol/mol to 100 μmol/mol*

---

## Author Comment (AC2) · 15 Oct 2020

*We thank Frank Pavia for his comments and suggestions on our manuscript. We agree with most comments and modified/updated the manuscript accordingly. Below is a point-by-point reply, [our answers appear in italics].*

**Frank Pavia:** General comments:
The manuscript by Roy-Barman et al. will be an excellent contribution to Biogeosciences, and to chemical oceanography in general. The work constitutes a novel way to estimate the mobility of mostly-insoluble elements during dust dissolution, and the findings are highly relevant to a bourgeoning literature on using Th isotopes to quantify dust dissolution and trace element inputs. The methods, data quality, and interpretations are generally sound. Before publication, there are a several points that need to be addressed, and I encourage the authors to reconsider alternate possibilities for their recommendations for the use of dissolved Th isotopes to quantify dust fluxes (see general comments below).

Specific comments:
1) I had a hard time following the REE section of the results (section 3.3.). I think this is because of the way Figure 2 is plotted. The sharp increase in REE at the start of the FAST experiment is difficult to see in Figure 2b because the t=0 and t=1 points look like they are at the same place on the x-axis. Maybe it would be good to label the t=1 point as being immediately after dust addition (if this is the case?).

*Reply: Points at t=0 (before dust addition for D and G series) are now labelled throughout the paper with crosses to be easily distinguished from points at t= 1h and latter (after dust addition for D and G series).*

**Frank Pavia:** It should also be clarified that the rapid increase in REE is only seen at FAST, and not at ION (unless I am having a hard time seeing this at ION because of the same plotting issue.

*Reply: At ION, we did not measure samples for t=0. The seawater composition without dust addition is given by the analysis of the "C"control series. Hence, there is also a sharp increase of the REE concentration at ION. This explanation is now given in the MS.*

**Frank Pavia:** This same point holds true for 232Th and Figure 3.
*Reply: For FAST, the points at t=0 are now indicated on this graph to. For ION, the comparison between the D (or G) series and the controls (C series without no dust amendment) highlight the sharp $^{232}$Th concentration increase after dust addition.*

**Frank Pavia:** Also, page 7 line 2 suggests that
230Th/232Th ratios as a function of time are plotted in Figure 3, but I do not see them there – I would quite like to see these ratios plotted with sampling time!

*Reply: The $^{230}$Th/$^{232}$Th ratio vs time is now show in the electronic supplement (Fig. ES3).*

**Frank Pavia:** 2) I'm curious from an analytical perspective, though not necessarily for the paper, where they think their high 231Pa blanks might come from, since there is so little 231Pa in nature. That said, the precision the authors achieved on such small quantities of 231Pa, in such small samples, is remarkable.

*Reply: Thank you. The high $^{231}Pa$ blanks came from an accidental combination of low chemistry yields for one sample set and high MC-ICPMS "machine blanks" (blank obtained by just running the dilute nitric+ HF solution used prepare the samples). At these very low signal level, it may not represent $^{231}Pa$, but result from formation of polyatomic ions desorbed from the MC-ICPMS source.*

**Frank Pavia:** 3) I am curious about the inferred dissolution of detrital calcium carbonate in the dust mentioned in 3.6. even though this isn't the main point of the paper. This process must be occurring above the calcite saturation horizon – what is driving this dissolution?

*Reply: The seeded dust was submitted to an evapo-condensation processing in lab in order to simulate atmospheric cloud processes, as specified in 2.1. During cloud processing simulation, the pH could be very low, enabling the dissolution of calcite present in the dust and a part of this calcium is re-precipitated as calcium carbonate (and even calcium hydrogen carbonate) during the evaporation step. Thus this calcium carbonate is not the "native" calcite in the dust. This is this carbonate which could be dissolved rapidly in the seawater.*

     *In order to be clearer on this point, we mention in 3.6. the effect of cloud processing: " The material collected in the traps contained 2.6% of Fe and 4.8% of Al (Tab. ES4). This is higher than the initial dust composition (2.3% of Fe and 3.3% of Al), due to preferential dissolution of highly soluble calcium carbonate or possibly calcium hydrogen carbonate formed during the simulation of dust processing in clouds (see section 2.1., Desboeufs et al., 2014).*

**Frank Pavia:** 4) I struggle with the arguments in the first paragraph of section 4 (page 7, particularly lines 34-41). What time interval are these waters seeded over? The dust input units given of 10g dust/m2 are not flux units without a time component, so it is difficult to assess whether this is a feasible dust flux compared to the highest dust pulses or mean annual dust fluxes. Because the dust corresponding to the average areal input was added in such a short time period, I think the sentence in lines 37-38 "Hence, the PEACETIME experiments: : :" must be taken out.

*Reply: We have removed the term of "flux" and replaced it by the term of "quantity". As noted by Frank Pavia, the flux implies a notion of time. In practice, during PEACETIME, dust was quickly sprayed over the four dust amended minicosms (successively, each 'rain' lasting ~ 20 mn) in order to be as synoptic as possible. This 20 mn "rain" time is now mentioned in section 2.1. This rather short duration is not unrealistic as this type of large, rapidly settling events have already been observed in the Mediterranean Sea, and are often associated with low rainfall (Loÿe-Pilot & Martin, 1996): as reported by these authors, "very significant amount of dust can be deposited with only few drops or mist", meaning that those events deposited high fluxes within only few minutes. However, what really matters for the experiment is the quantity of dust deposited rather than how fast it was deposited.*

*Ref: Loÿe-Pilot, M. D., & Martin, J. M. (1996). Saharan dust input to the western Mediterranean: an eleven years record in Corsica. In The impact of desert dust across the Mediterranean (pp. 191-199). Springer, Dordrecht.*

**Frank Pavia:** 5) The difference in dissolution kinetics between Th/REE and Al is intriguing, and I would like to have seen the authors speculate on what drives it. Are different mineral phases releasing Th/REE than Al?

*Reply: we now mention in section 4.1. that an unexpected result of the PEACETIME experiments is the contrasting dissolution kinetics of Al relative to Th and REE. Th and REE are (at least partly) carried by specific REE and Th rich phases (Marchandise et al., 2014), that may partly account for the decoupling with Al. Alternatively, the fast dissolution of calcium carbonate or calcium hydrogen carbonate formed during the cloud simulation step could account for REE and Th release (see sections 3.6 and 4.3.).*

**Frank Pavia:** A back of the envelope calculation, 3.6g dust added, 15% of it presumed to be CaCO3 that dissolves, 0.5ppm hydrogenous 232Th in lacustrine carbonates (Lin et al., 1996) and in pedogenic carbonates (Ludwig and Paces, 2002) gives 1160 pmol of 232Th that could be released into the tank from detrital CaCO3 dissolution. Dividing this by the 300L tank volume gives a _[232Th] of _3.9pM – very close to the 3-5 pM initial rapid increase in 232Th observed at both ION and FAST. This calculation should be done by the authors for REE as well. If this is the case, then Th dissolution in dust may be highly dependent on the detrital carbonate content – which would be a really interesting result. Regardless, it is important to explicitly note that Th and REE release from dust is clearly controlled by multiple phases – a fast dissolving one and a slow-dissolving one – and that this is different from Fe and Al release.

*Reply: We have included the calculation suggested by Frank Pavia, but we have used references concerning carbonates from the Sahara:*
*Alternatively, the dissolution of the carbonates from the dusts can release significant amounts of $^{232}$Th. For example, the $^{232}$Th content of travertine and pedogenic carbonates found in the Western Sahara ranges from 0.5 to 12 ppm (Szabo et al., 1995, Candy et al., 2004, Weisrock et al., 2008). Taking 2 ppm as mean value and considering that carbonate dissolution represents 4.5 % of the dust mass (see section 3.6), it corresponds to a release of 1400 pmol in 300L of seawater potentially yielding an increase of 5 pM, in gross agreement with observations (Fig. 3). While pedogenic calcretes contain sufficient amounts of $^{232}$Th and REE to account for the changes of Th and REE concentrations observed during the Peacetime experiments (Prudencio et al., 2011), the $^{230}$Th /$^{232}$Th ratio of these carbonates is generally low ($^{230}$Th /$^{232}$Th = 2-5×10-6, Candy et al., 2004), so that it cannot account for the higher $^{230}$Th /$^{232}$Th ratio (~ 8×10-6) of the Th released during the PEACETIME experiments (Fig. 6).*

*Candy, I., Black, S., & Sellwood, B. W.: Quantifying time scales of pedogenic calcrete formation using U-series disequilibria. Sedimentary Geology, 170(3-4), 177-187, 2004.*

*Szabo, B. J., Haynes Jr, C. V., & Maxwell, T. A.: Ages of Quaternary pluvial episodes determined by uranium-series and radiocarbon dating of lacustrine deposits of Eastern Sahara. Palaeogeography palaeoclimatology palaeoecology, 1995.*

*Prudêncio, M. I., Dias, M. I., Waerenborgh, J. C., Ruiz, F., Trindade, M. J., Abad, M., ... & Gouveia, M. A.: Rare earth and other trace and major elemental distribution in a pedogenic calcrete profile (Slimene, NE Tunisia). Catena, 87, 147-156, 2011.*

*Weisrock, A., Rousseau, L., Reyss, J. L., Falguères, C., Ghaleb, B., Bahain, J. J., ... & Pozzi, J. P.: Travertines of the Moroccan Sahara northern border: morphological settings, U-series datings and palaeoclimatic indications. Géomorphologie: relief, processus, environnement, (3), 153, 2008.*

**Frank Pavia:** 6) The relative fractional solubility of metals is used in calculations of dissolved metal flux from dust derived from thorium isotopes (Hayes et al., 2018; Pavia et al., 2020).

It would be fantastic to calculate the release ratios of Fe, Al, and REE to Th (along with their uncertainties) and provide these in addition to Table 1. These would be an excellent addition to the aerosol-based solubility estimates currently used, though it would also be important to note that using these relative solubilities in Th-based Fe flux estimates could be biased if the differences in metal release kinetics between Fe and Th release are sufficiently large to decouple the release depths of Fe and Th as fine lithogenic particles aggregate and sink from surface waters.

***Reply:*** *The release ratios of Fe, Al, and REE to Th (along with their uncertainties) are now given in Table ES5 and discussed in section 4.4.*

**Frank Pavia:** 7) There is a mismatch by a factor of 10 between the units on y-axis of Figure 6 and those cited in the text in section 4.3.

***Reply:*** *The factor 10 is now corrected*

**Frank Pavia**: 8) The points regarding differences in solubility estimates of Th between previous results and this study made in 4.4. are very important. Th solubility is one of the largest sources of uncertainty in Th-based dust flux estimates (Pavia et al., 2020), and different methods give different results. However, authors' points about lateral 232Th inputs are not justified by the data presented in this paper. The solubility of 232Th does not alter the removal residence time of Th in the upper water column. The 230Th-based removal residence times in the upper 250m of the South Pacific Gyre (depths over which Th-based dust flux estimates are integrated) are 1-2 years (Pavia et al., 2020).

Note that the South Pacific Gyre is highly oligotrophic, similar to the Mediterranean – so 1-2 years is likely on the longer end of Th residence times for the upper 250m in the ocean. The diffusive lengthscale for lateral diffusion of Th in the upper water column, given by $_x=_p (2K_H \_t)$ (Roy-Barman, 2009), for a residence time of 2 years and a lateral eddy diffusivity of 10^3 m2/s, is 355km. So, regardless of the 232Th solubility, lateral lithogenic inputs from the continents should not affect 232Th distributions in the open ocean beyond this distance. Indeed, the lack of lateral inputs affecting the deep Mediterranean, with longer transport timescales from boundary lithogenic sources than in the upper water column in the open ocean, is noted in the conclusion. More work is needed to determine the role of vertical and lateral exchange on 230Th and 232Th budgets, but the conclusions of this study do not justify the conclusiveness of the statements regarding lateral Th fluxes (e.g. "cannot be done", page 12 line 33; "strongly biased by advective inputs" page 12 lines 31-32). These statements should be tempered accordingly. Similarly, the statement on page 13 lines 13-15 should be

adjusted or removed, as the current study does not provide any insights on the importance of advection as a source of 232Th to the open ocean.

*Reply: The sentences raising possibilities of bias in the estimation of Fe and Th solubilities based on the 230Th-232Th method have been tempered or removed. However, we note that the equation relating the time and length during eddy diffusion must be used with care for 2 reasons:*
*- It neglects advection. For example, this equation does not account for the Fukushima radionuclide transport across the North Pacific in 2 years (Buesseler et al., 2016).*
*- the net effect of diffusion depends on both $K_h$ and the concentration gradient.*
*$F = - K_h \, d^{232}Th/dz$*
*When large gradients occur (coastal ocean/open ocean), it may take more than one diffusion length to totally remove/dilute the high coastal concentrations and reach constant concentration in the open ocean. This is why we add in section 4.4 that "Hence, deducing dust inputs from the $^{230}Th/^{232}Th$ ratio of surface waters (Hsieh et al., 2012) requires first to consider ocean areas where the water residence time relative to circulation significantly exceeds the Th residence time relative to scavenging (e.g.: part of the south Pacific gyre where the horizontal dissolved $^{232}Th$ gradient tend to vanish (Pavia et al., 2020))."*

*Buesseler, K., Dai, M., Aoyama, M., Benitez-Nelson, C., Charmasson, S., Higley, K., ... & Smith, J. N. (2017). Fukushima Daiichi–derived radionuclides in the ocean: transport, fate, and impacts. Annual review of marine science, 9, 173-203.*

---

## Author Response (AR1)

Dear Jan-Berend Stuut,

Please find below the revised and marked-up version of the manuscript.

Best regards.

Matthieu Roy-Barman

[revised manuscript text omitted]

Fig 1

[b24]

[Figure]

Fig 2a.

[Figure]

15 Fig 2b.[b25]

[Figure]

Fig. 3[b26]

[Figure]

15 Fig. 4: [b27]

[Figure]

Fig. 5.

[Figure]

Fig. 6 [b28]

Tab. ES1: REE and Th GEOTRACES standard analyses

| | light REE | | | | | | | | medium REE | | | | | | | | | | heavy REE | | | | | | | | | | Thorium | | | |
|---|---|---|---|---|---|---|---|---|---|---|---|---|---|---|---|---|---|---|---|---|---|---|---|---|---|---|---|---|---|---|---|---|
| | La | 2σ | Ce | 2σ | Pr | 2σ | Nd | 2σ | Sm | 2σ | Eu | 2σ | Gd | 2σ | Tb | 2σ | Dy | 2σ | Ho | 2σ | Er | 2σ | Tm | 2σ | Yb | 2σ | Lu | 2σ | 232Th | 2σ | 230Th | 2σ |
| | (pM) | | (pM) | | (pM) | | (pM) | | (pM) | | (pM) | | (pM) | | (pM) | | (pM) | | (pM) | | (pM) | | (pM) | | (pM) | | (pM) | | (pM) | | (aM) | |
| BATS 2000 m_1 | 17.9 | 0.47 | 5.37 | 0.2 | 3.84 | 0.12 | 17.2 | 0.5 | 3.56 | 0.2 | 0.89 | 0.05 | 4.9 | 0.2 | 0.76 | 0.03 | 5.55 | 0.2 | 1.41 | 0.1 | 4.66 | 0.2 | 0.69 | 0.02 | 4.5 | 0.2 | 0.72 | 0.03 | 113 | 160 | 37.3 | 1.1 |
| BATS 2000 m_2 | 16.9 | 0.35 | 4.86 | 0.1 | 3.77 | 0.12 | 17.22 | 0.4 | 3.57 | 0.2 | 0.91 | 0.06 | 4.7 | 0.2 | 0.75 | 0.03 | 5.65 | 0.2 | 1.4 | 0.0 | 4.66 | 0.1 | 0.68 | 0.03 | 4.42 | 3.43 | 0.69 | 0.02 | 140 | 165 | 36.1 | 2.1 |
| BATS 2000 m_3 | 16.4 | 0.84 | 4.94 | 0.2 | 3.72 | 0.13 | 16.96 | 0.5 | 3.58 | 0.2 | 0.89 | 0.04 | 4.5 | 0.2 | 0.76 | 0.03 | 5.5 | 0.2 | 1.36 | 0.1 | 4.62 | 0.2 | 0.64 | 0.04 | 4.33 | 0.48 | 0.69 | 0.05 | 126 | 171 | 37.8 | 0.9 |
| average | 17.1 | 1.08 | 5.06 | 0.4 | 3.77 | 0.08 | 17.12 | 0.2 | 3.57 | 0.02 | 0.9 | 0.01 | 4.7 | 0.3 | 0.76 | 0.01 | 5.57 | 0.1 | 1.39 | 0.0 | 4.65 | 0.04 | 0.67 | 0.04 | 4.41 | 0.12 | 0.7 | 0.03 | 126 | 19 | 37 | 1 |
| | | | | | | | | | | | | | | | | | | | | | | | | | | | | | | | | |
| consensual value | 23 | | 5.0 | | 3.9 | | 16.9 | | 3.4 | | 0.9 | | 4.7 | | 0.8 | | 5.7 | | 1.5 | | 4.9 | | 0.7 | | 4.6 | | 0.8 | | 208 | | 38 | |
| | 2.7 | | 2.2 | | 0.3 | | 1.2 | | 0.3 | | 0.1 | | 0.5 | | 0.1 | | 0.4 | | 0.1 | | 0.2 | | 0.0 | | 0.2 | | 0.0 | | 42 | | 6 | |

Tab. ES2: Dissolved Fe and Al data

| Station | tank | time (h) | DFe (nM) | DAl (nM) | Station | tank | time (h) | DFe (nM) | DAl (nM) | Station | tank | time (h) | DFe (nM) | DAl (nM) |
|---|---|---|---|---|---|---|---|---|---|---|---|---|---|---|
| TYR | C1 | 0 | 1.54 | 46.3 | ION | C1 | 0 | 2.49 | 69.6 | FAST | C1 | 0 | 1.73 | 24.1 |
|  | C1 | 24 | 1.11 | 47.4 |  | C1 | 24 | 1.60 | 80.3 |  | C1 | 24 | 1.34 | 23.5 |
|  | C1 | 72 | 1.60 | 47.8 |  | C1 | 72 | 1.37 | 81.1 |  | C1 | 72 | 0.88 | 27.4 |
|  |  |  |  |  |  |  |  |  |  |  |  |  |  |  |
|  | C2 | 0 | 1.53 | 50.3 |  | C2 | 0 | 2.49 | 79.6 |  | C2 | 0 | 1.94 | 24 |
|  | C2 | 24 | 0.67 | 42.8 |  | C2 | 24 | 1.78 | 80.2 |  | C2 | 24 | 1.80 | 23.5 |
|  | C2 | 72 | 1.41 | 46.1 |  | C2 | 72 | 1.55 | 71 |  | C2 | 72 | 0.99 | 27.7 |
|  |  |  |  |  |  |  |  |  |  |  |  |  |  |  |
|  | D1 | 0 | 1.46 | 43.9 |  | D1 | 0 | 2.84 | 70.7 |  | D1 | 0 | 6.68 | 49 |
|  | D1 | 24 | 1.45 | 79 |  | D1 | 24 | 1.64 | 115.7 |  | D1 | 24 | 2.66 | 61 |
|  | D1 | 72 | 1.35 | 102.2 |  | D1 | 72 | 1.64 | 138.4 |  | D1 | 96 | 9.69 | 107.4 |
|  |  |  |  |  |  |  |  |  |  |  |  |  |  |  |
|  | D2 | 0 | 1.61 | 44.1 |  | D2 | 0 | NA | 70.7 |  | D2 | 0 | 6.14 | 22.4 |
|  | D2 | 24 | 0.53 | 81.2 |  | D2 | 24 | 1.52 | 113.4 |  | D2 | 24 | 3.93 | 62.7 |
|  | D2 | 72 | 0.87 | 98.7 |  | D2 | 72 | 1.36 | 135.6 |  | D2 | 96 | 2.01 | 99.3 |
|  |  |  |  |  |  |  |  |  |  |  |  |  |  |  |
|  | G1 | 0 | 1.73 | 45.4 |  | G1 | 0 | 5.10 | 83 |  | G1 | 0 | 3.05 | 25.4 |
|  | G1 | 24 | 0.71 | 86.6 |  | G1 | 24 | 2.52 | 112.7 |  | G1 | 24 | 2.29 | 62.7 |
|  | G1 | 72 | 3.21 | 109.9 |  | G1 | 72 | 3.53 | 136.4 |  | G1 | 96 | 2.85 | 107.7 |
|  |  |  |  |  |  |  |  |  |  |  |  |  |  |  |
|  | G2 | 0 | 1.13 | 48.8 |  | G2 | 0 | 2.04 | 79.3 |  | G2 | 0 | 1.59 | 23.6 |
|  | G2 | 24 | 1.05 | 76 |  | G2 | 24 | 1.53 | 112.5 |  | G2 | 24 | 3.89 | 60.5 |
|  | G2 | 72 | 0.90 | 101.3 |  | G2 | 72 | 1.45 | 144.2 |  | G2 | 96 | 1.16 | 104.3 |

Tab. ES3: Dissolved REE, Th and Pa data

| tank | time | La | 2σ | Ce | 2σ | Pr | 2σ | Nd | 2σ | Sm | 2σ | Eu | 2σ | Gd | 2σ | Tb | 2σ | Dy | 2σ | Ho | 2σ | Er | 2σ | Tm | 2σ | Yb | 2σ | Lu | 2σ | 232Th | 2σ | 230Th | 2σ | 231Pa | 2σ |
|---|---|---|---|---|---|---|---|---|---|---|---|---|---|---|---|---|---|---|---|---|---|---|---|---|---|---|---|---|---|---|---|---|---|---|---|
| | (h) | (pM) | | (pM) | | (pM) | | (pM) | | (pM) | | (pM) | | (pM) | | (pM) | | (pM) | | (pM) | | (pM) | | (pM) | | (pM) | | (pM) | | (pM) | | (aM) | | (aM) | |
| ION C1 | 1 | 21 | 1 | 22 | 1 | 5.6 | 0.2 | 25.7 | 0.6 | 6.1 | 0.3 | 0.0 | 0.1 | 8.5 | 0.5 | 1.4 | 0.1 | 10.3 | 0.5 | 2.4 | 0.2 | 7.9 | 0.4 | 1.1 | 0.1 | 7.2 | 0.3 | 1.1 | 0.0 | 1.0 | 0.1 | 11.4 | 0.9 | | |
| ION C1 | 24 | 29 | 1 | 36 | 1 | 7.6 | 0.3 | 33.9 | 0.6 | 7.7 | 0.4 | 2.0 | 0.1 | 10.2 | 0.4 | 1.6 | 0.1 | 11.3 | 0.5 | 2.6 | 0.1 | 8.5 | 0.3 | 1.1 | 0.1 | 7.5 | 0.2 | 1.2 | 0.1 | 1.2 | 0.1 | 13.9 | 1.0 | | |
| ION C1 | 72 | 28 | 1 | 35 | 1 | 7.1 | 0.3 | 32.1 | 0.9 | 7.7 | 0.4 | 2.0 | 0.1 | 9.8 | 0.4 | 1.6 | 0.1 | 11.3 | 0.4 | 2.6 | 0.1 | 8.3 | 0.4 | 1.2 | 0.1 | 7.5 | 0.2 | 1.1 | 0.0 | 1.2 | 0.1 | 10.9 | 1.0 | | |
| ION C2 | 1 | 22 | 1 | 22 | 1 | 5.7 | 0.2 | 25.9 | 0.8 | 6.2 | 0.4 | 1.6 | 0.1 | 8.5 | 0.4 | 1.4 | 0.1 | 10.2 | 0.5 | 2.5 | 0.1 | 8.1 | 0.4 | 1.1 | 0.0 | 7.3 | 0.3 | 1.1 | 0.1 | 0.9 | 0.1 | 11.5 | 0.9 | | |
| ION C2 | 24 | 24 | 1 | 29 | 1 | 6.4 | 0.2 | 29.5 | 0.4 | 6.8 | 0.2 | 1.8 | 0.1 | 8.9 | 0.3 | 1.5 | 0.1 | 10.6 | 0.4 | 2.5 | 0.1 | 8.1 | 0.3 | 1.1 | 0.0 | 7.5 | 0.3 | 1.1 | 0.1 | 1.0 | 0.1 | 10.1 | 0.6 | | |
| ION C2 | 72 | 25 | 1 | 30 | 1 | 6.4 | 0.3 | 29.0 | 0.5 | 6.8 | 0.3 | 1.7 | 0.1 | 9.0 | 0.4 | 1.5 | 0.1 | 10.7 | 0.3 | 2.5 | 0.1 | 8.1 | 0.3 | 1.1 | 0.1 | 7.5 | 0.2 | 1.1 | 0.1 | 1.0 | 0.1 | 10.3 | 0.8 | | |
| ION D1 | 1 | 75 | 4 | 144 | 7 | 23.8 | 1.1 | 107.2 | 2.0 | 24.6 | 1.3 | 5.8 | 0.4 | 26.8 | 1.3 | 4.0 | 0.2 | 25.4 | 1.1 | 5.1 | 0.2 | 15.5 | 0.8 | 2.0 | 0.1 | 12.7 | 0.4 | 1.8 | 0.1 | 4.9 | 0.1 | 47.3 | 1.3 | | |
| ION D1 | 24 | 72 | 3 | 141 | 5 | 21.2 | 0.6 | 95.6 | 2.5 | 21.2 | 0.7 | 5.1 | 0.2 | 24.9 | 0.8 | 3.7 | 0.1 | 24.0 | 0.8 | 5.0 | 0.2 | 15.4 | 0.5 | 2.0 | 0.1 | 12.6 | 0.3 | 1.9 | 0.1 | 3.7 | 0.1 | 32.7 | 1.8 | | |
| ION D1 | 72 | 60 | 2 | 122 | 5 | 17.9 | 0.6 | 80.5 | 2.1 | 18.2 | 0.7 | 4.5 | 0.2 | 22.5 | 0.8 | 3.4 | 0.2 | 22.8 | 0.9 | 4.8 | 0.2 | 14.8 | 0.5 | 1.9 | 0.1 | 12.4 | 0.7 | 1.8 | 0.1 | 2.8 | 0.1 | 24.9 | 1.1 | | |
| ION D2 | 1 | 72 | 2 | 140 | 5 | 23.3 | 0.8 | 108.4 | 1.8 | 24.2 | 0.9 | 5.7 | 0.2 | 27.2 | 1.1 | 4.0 | 0.2 | 25.4 | 1.2 | 5.2 | 0.2 | 15.4 | 0.6 | 2.0 | 0.1 | 12.8 | 0.7 | 1.9 | 0.1 | 5.1 | 0.1 | 46.5 | 1.2 | | |
| ION D2 | 24 | 70 | 3 | 137 | 4 | 21.0 | 0.8 | 94.2 | 1.5 | 21.5 | 1.0 | 5.2 | 0.3 | 24.9 | 0.9 | 3.8 | 0.1 | 24.3 | 1.1 | 5.1 | 0.3 | 15.4 | 0.7 | 2.0 | 0.1 | 12.7 | 0.3 | 1.8 | 0.1 | 3.9 | 0.1 | 37.1 | 1.7 | | |
| ION D2 | 72 | 66 | 2 | 132 | 4 | 19.5 | 0.8 | 88.9 | 0.9 | 20.0 | 0.8 | 4.8 | 0.2 | 24.0 | 1.0 | 3.7 | 0.2 | 24.3 | 1.4 | 5.1 | 0.2 | 15.8 | 0.7 | 2.1 | 0.1 | 13.2 | 0.7 | 1.9 | 0.1 | 3.0 | 0.1 | 27.9 | 2.2 | | |
| ION G1 | 1 | 78 | 3 | 150 | 6 | 23.7 | 0.9 | 106.9 | 1.4 | 23.6 | 1.2 | 5.6 | 0.3 | 26.5 | 0.9 | 3.9 | 0.1 | 24.2 | 1.1 | 5.0 | 0.3 | 14.7 | 0.5 | 1.9 | 0.1 | 12.1 | 0.4 | 1.7 | 0.1 | 3.5 | 0.1 | 33.1 | 1.5 | | |
| ION G1 | 24 | 79 | 3 | 151 | 7 | 22.1 | 1.0 | 98.7 | 1.6 | 21.4 | 1.0 | 5.1 | 0.2 | 24.8 | 1.2 | 3.7 | 0.2 | 23.5 | 0.9 | 4.9 | 0.2 | 14.7 | 0.5 | 1.9 | 0.1 | 12.0 | 0.7 | 1.8 | 0.1 | 2.3 | 0.1 | 22.4 | 1.2 | | |
| ION G1 | 72 | 73 | 3 | 142 | 7 | 19.7 | 0.8 | 88.0 | 1.4 | 19.3 | 1.1 | 4.6 | 0.2 | 23.3 | 1.1 | 3.5 | 0.2 | 22.5 | 0.9 | 4.8 | 0.2 | 14.3 | 0.6 | 1.9 | 0.1 | 12.0 | 0.4 | 1.7 | 0.1 | 2.0 | 0.1 | 18.0 | 0.9 | | |
| ION G2 | 1 | 82 | 3 | 156 | 5 | 24.3 | 0.8 | 109.2 | 2.5 | 24.2 | 1.0 | 5.7 | 0.3 | 27.2 | 1.2 | 3.9 | 0.2 | 24.7 | 1.2 | 5.1 | 0.2 | 15.0 | 0.7 | 2.0 | 0.1 | 12.1 | 0.3 | 1.8 | 0.1 | 3.6 | 0.1 | 34.5 | 3.4 | | |
| ION G2 | 24 | 80 | 3 | 154 | 7 | 22.4 | 0.9 | 101.1 | 2.2 | 21.8 | 1.0 | 5.2 | 0.3 | 25.4 | 1.2 | 3.8 | 0.2 | 23.8 | 1.0 | 5.0 | 0.2 | 14.8 | 0.5 | 2.0 | 0.1 | 12.4 | 0.5 | 1.8 | 0.1 | 2.3 | 0.1 | 19.7 | 0.8 | | |
| ION G2 | 72 | 72 | 2 | 141 | 4 | 19.3 | 0.7 | 85.6 | 2.5 | 18.7 | 0.7 | 4.6 | 0.2 | 22.4 | 0.8 | 3.4 | 0.1 | 22.2 | 0.8 | 4.8 | 0.1 | 14.4 | 0.5 | 1.9 | 0.1 | 12.0 | 0.3 | 1.8 | 0.1 | 2.0 | 0.1 | 17.3 | 1.1 | | |
| FA C1 | 0 | 19 | 1 | 20 | 1 | 5.1 | 0.2 | 23.6 | 1.1 | 5.9 | 0.4 | 1.5 | 0.1 | 8.1 | 0.4 | 1.2 | 0.1 | 8.1 | 0.4 | 1.9 | 0.1 | 6.1 | 0.3 | 0.8 | 0.1 | 5.1 | 0.3 | 0.8 | 0.0 | 1.1 | 0.1 | 11.8 | 1.3 | 2.6 | 0.5 |
| FA C1 | 1 | 20 | 1 | 23 | 1 | 5.4 | 0.2 | 26.1 | 1.0 | 6.2 | 0.4 | 1.6 | 0.1 | 8.7 | 0.4 | 1.3 | 0.1 | 8.6 | 0.4 | 2.0 | 0.1 | 6.3 | 0.3 | 0.8 | 0.1 | 5.3 | 0.2 | 0.8 | 0.1 | 1.3 | 0.1 | 16.4 | 1.5 | 2.6 | 0.5 |
| FA C1 | 6 | 24 | 1 | 35 | 1 | 6.9 | 0.3 | 32.6 | 1.2 | 7.7 | 0.4 | 1.9 | 0.1 | 10.5 | 0.5 | 1.5 | 0.1 | 9.8 | 0.5 | 2.1 | 0.1 | 6.8 | 0.4 | 0.9 | 0.1 | 5.7 | 0.3 | 0.9 | 0.1 | 1.4 | 0.0 | 20.4 | 1.2 | 3.3 | 0.5 |
| FA C1 | 12 | 32 | 1 | 58 | 1 | 9.6 | 0.4 | 43.1 | 1.8 | 10.2 | 0.5 | 2.5 | 0.1 | 13.4 | 0.6 | 1.7 | 0.1 | 10.9 | 0.5 | 2.3 | 0.1 | 7.0 | 0.3 | 0.9 | 0.0 | 6.0 | 0.2 | 0.9 | 0.0 | 10.9 | 0.1 | 94.7 | 2.5 | 2.5 | 0.5 |
| FA C1 | 24 | 26 | 1 | 35 | 1 | 6.9 | 0.3 | 32.6 | 1.3 | 8.0 | 0.5 | 1.9 | 0.2 | 11.2 | 0.7 | 1.5 | 0.1 | 10.1 | 0.7 | 2.2 | 0.1 | 6.9 | 0.4 | 0.9 | 0.1 | 5.9 | 0.4 | 0.9 | 0.1 | 1.5 | 0.1 | 28.4 | 1.8 | 1.8 | 0.7 |
| FA C1 | 48 | 29 | 1 | 38 | 2 | 7.2 | 0.3 | 33.3 | 1.3 | 8.0 | 0.4 | 1.9 | 0.1 | 11.1 | 0.6 | 1.4 | 0.1 | 9.6 | 0.5 | 2.1 | 0.1 | 6.7 | 0.4 | 0.9 | 0.1 | 5.6 | 0.2 | 1.2 | 0.1 | 1.1 | 0.1 | 10.9 | 1.3 | 2.4 | 0.5 |
| FA C1 | 72 | 29 | 1 | 33 | 1 | 6.6 | 0.3 | 29.5 | 1.2 | 7.2 | 0.4 | 1.7 | 0.1 | 10.1 | 0.6 | 1.3 | 0.1 | 9.4 | 0.5 | 2.1 | 0.1 | 6.7 | 0.4 | 0.9 | 0.1 | 5.7 | 0.3 | 0.9 | 0.1 | 1.1 | 0.1 | 14.1 | 1.8 | 2.7 | 0.6 |
| FA C1 | 96 | 30 | 1 | 34 | 1 | 6.7 | 0.3 | 31.2 | 1.2 | 7.5 | 0.5 | 1.8 | 0.1 | 10.6 | 0.6 | 1.4 | 0.1 | 9.5 | 0.5 | 2.1 | 0.1 | 6.8 | 0.4 | 0.9 | 0.1 | 5.7 | 0.3 | 0.9 | 0.1 | 1.1 | 0.1 | 17.9 | 1.8 | 2.3 | 0.5 |
| FA C2 | 12 | 25 | 2 | 29 | 2 | 6.2 | 0.6 | 28.4 | 0.7 | 6.7 | 0.6 | 1.8 | 0.1 | 8.4 | 0.7 | 1.3 | 0.1 | 9 | 0.8 | 2.0 | 0.1 | 6.4 | 0.5 | 0.8 | 0.1 | 5.3 | 0.2 | 0.8 | 0.1 | ND | ND | 0.0 | 0.0 | | |
| FA C2 | 48 | 28 | 1 | 35 | 1 | 6.6 | 0.4 | 30.5 | 0.3 | 7.2 | 0.4 | 1.9 | 0.1 | 8.8 | 0.6 | 1.4 | 0.1 | 9 | 0.5 | 2.0 | 0.1 | 6.6 | 0.3 | 0.9 | 0.0 | 5.5 | 0.3 | 0.8 | 0.0 | 1.5 | 0.8 | 26.4 | 17 | | |
| FA C2 | 96 | 29 | 1 | 32 | 1 | 6.2 | 0.2 | 28.9 | 0.6 | 6.8 | 0.5 | 1.8 | 0.1 | 8.6 | 0.3 | 1.3 | 0.1 | 9 | 0.4 | 2.1 | 0.1 | 6.6 | 0.3 | 0.9 | 0.1 | 5.4 | 0.2 | 0.8 | 0.1 | 0.7 | 0.1 | 4.9 | 0.8 | | |
| FA D1 | 0 | 22 | 1 | 27 | 1 | 5.9 | 0.3 | 27.6 | 1.2 | 6.4 | 0.4 | 1.6 | 0.1 | 9.2 | 0.6 | 1.3 | 0.1 | 9 | 0.5 | 2.0 | 0.1 | 6.1 | 0.4 | 0.8 | 0.1 | 5.3 | 0.3 | 0.8 | 0.1 | 1.5 | 0.0 | 18.0 | 1.2 | 2.1 | 0.5 |
| FA D1 | 1 | 71 | 3 | 151 | 6 | 24.4 | 1.0 | 110.8 | 4.0 | 25.2 | 1.0 | 6.0 | 0.3 | 33.1 | 1.4 | 4.1 | 0.2 | 25 | 1.0 | 4.8 | 0.2 | 13.7 | 0.6 | 1.7 | 0.1 | 10.8 | 0.4 | 1.5 | 0.1 | 5.9 | 0.1 | 65.5 | 3.0 | 2.7 | 0.4 |
| FA D1 | 6 | 76 | 3 | 149 | 6 | 23.4 | 1.0 | 106.6 | 4.3 | 24.1 | 1.5 | 5.8 | 0.3 | 32.4 | 1.5 | 4.0 | 0.2 | 25 | 1.1 | 4.8 | 0.2 | 14.1 | 0.6 | 1.8 | 0.1 | 11.1 | 0.4 | 1.5 | 0.1 | 5.4 | 0.1 | 57.5 | 2.0 | 2.4 | 0.6 |
| FA D1 | 12 | 75 | 2 | 151 | 6 | 23.1 | 0.9 | 104.6 | 4.1 | 23.3 | 1.1 | 5.6 | 0.3 | 32.3 | 1.7 | 3.9 | 0.2 | 24 | 1.4 | 4.8 | 0.3 | 14.0 | 0.9 | 1.8 | 0.1 | 11.1 | 0.7 | 1.6 | 0.1 | 10.2 | 0.1 | 85.4 | 2.4 | 2.7 | 0.4 |
| FA D1 | 24 | 73 | 3 | 149 | 6 | 22.3 | 1.0 | 100.3 | 4.3 | 22.8 | 1.3 | 5.3 | 0.3 | 30.9 | 1.6 | 3.8 | 0.2 | 24 | 1.3 | 4.7 | 0.3 | 13.8 | 0.8 | 1.7 | 0.1 | 11.1 | 0.7 | 1.6 | 0.1 | 9.3 | 0.1 | 72.7 | 2.0 | 2.2 | 0.5 |
| FA D1 | 48 | 58 | 3 | 127 | 6 | 19.2 | 0.8 | 89.4 | 3.7 | 20.2 | 1.1 | 5.0 | 0.2 | 28.6 | 1.3 | 3.7 | 0.2 | 23 | 1.1 | 4.8 | 0.2 | 14.2 | 0.7 | 1.8 | 0.1 | 11.4 | 0.4 | 1.6 | 0.1 | 3.0 | 0.1 | 25.1 | 1.9 | 2.0 | 0.8 |
| FA D1 | 72 | 53 | 2 | 110 | 5 | 16.9 | 0.8 | 79.2 | 3.5 | 18.0 | 1.0 | 4.4 | 0.3 | 25.6 | 1.4 | 3.2 | 0.2 | 21 | 1.1 | 4.4 | 0.3 | 13.1 | 0.7 | 1.7 | 0.1 | 10.9 | 0.6 | 1.6 | 0.1 | 2.3 | 0.1 | 32.0 | 1.7 | 2.4 | 0.5 |
| FA D1 | 96 | 58 | 2 | 97 | 3 | 15.5 | 0.6 | 70.7 | 2.7 | 16.7 | 1.1 | 3.9 | 0.3 | 23.4 | 1.4 | 3.0 | 0.2 | 20 | 1.3 | 4.3 | 0.3 | 12.8 | 0.8 | 1.6 | 0.1 | 9.9 | 0.6 | 1.6 | 0.1 | 2.1 | 0.1 | 44.8 | 2.4 | 2.7 | 0.6 |
| FA D2 | 1 | 20 | 3 | 23 | 4 | 5.3 | 0.9 | 24.8 | 0.4 | 6.1 | 1.0 | 1.5 | 0.3 | 7.6 | 1.2 | 1.2 | 0.2 | 8 | 1.4 | 1.9 | 0.3 | 5.9 | 1.0 | 0.8 | 0.1 | 5.0 | 0.3 | 0.8 | 0.1 | 1.0 | 0.1 | 8.8 | 1.0 | | |
| FA D2 | 48 | 65 | 2 | 130 | 4 | 24.2 | 0.7 | 107.3 | 1.2 | 20.2 | 0.8 | 5.0 | 0.2 | 24.8 | 0.7 | 3.6 | 0.1 | 23 | 0.9 | 4.7 | 0.2 | 13.8 | 0.5 | 1.7 | 0.1 | 10.7 | 0.3 | 1.5 | 0.1 | 3.0 | 0.1 | 28.7 | 1.3 | | |
| FA D2 | 96 | 57 | 2 | 113 | 4 | 17.5 | 0.5 | 81.0 | 1.2 | 18.0 | 0.8 | 4.3 | 0.2 | 21.5 | 0.8 | 3.2 | 0.2 | 21 | 1.1 | 4.4 | 0.2 | 13.2 | 0.7 | 1.7 | 0.1 | 10.5 | 0.3 | 1.6 | 0.1 | 2.3 | 0.1 | 20.1 | 1.3 | | |
| FA G1 | 1 | 83 | 2 | 163 | 5 | 25.4 | 0.8 | 115.1 | 0.9 | 25.2 | 0.8 | 5.7 | 0.2 | 27.6 | 1.1 | 3.9 | 0.1 | 24 | 0.6 | 4.8 | 0.2 | 13.7 | 0.5 | 1.7 | 0.1 | 10.6 | 0.5 | 1.5 | 0.0 | 4.8 | 0.3 | 40.6 | 2.1 | | |
| FA G1 | 48 | 69 | 4 | 139 | 10 | 21.2 | 1.7 | 98.3 | 2.8 | 21.2 | 1.8 | 5.1 | 0.4 | 25.3 | 2.0 | 3.6 | 0.2 | 23 | 1.7 | 4.7 | 0.4 | 13.7 | 1.1 | 1.8 | 0.1 | 10.9 | 0.4 | 1.6 | 0.1 | 1.8 | 0.1 | 18.9 | 1.5 | | |
| FA G1 | 96 | 69 | 2 | 127 | 4 | 19.4 | 0.6 | 86.0 | 1.9 | 19.3 | 0.8 | 4.6 | 0.1 | 22.6 | 0.6 | 3.4 | 0.1 | 22 | 1.0 | 4.6 | 0.1 | 13.4 | 0.3 | 1.8 | 0.0 | 10.7 | 0.5 | 1.6 | 0.0 | 1.5 | 0.1 | 12.9 | 1.3 | | |
| FA G2 | 1 | 78 | 3 | 162 | 7 | 25.4 | 1.0 | 115.5 | 3.3 | 25.3 | 1.2 | 5.8 | 0.2 | 27.4 | 1.1 | 3.9 | 0.2 | 23 | 0.9 | 4.7 | 0.2 | 13.3 | 0.6 | 1.7 | 0.1 | 10.3 | 0.7 | 1.4 | 0.1 | 4.7 | 0.1 | 40.5 | 1.7 | | |
| FA G2 | 48 | 70 | 3 | 140 | 5 | 20.8 | 0.7 | 94.7 | 4.2 | 20.5 | 0.9 | 4.9 | 0.2 | 24.0 | 1.3 | 3.5 | 0.2 | 22 | 1.0 | 4.5 | 0.3 | 13.5 | 0.6 | 1.7 | 0.1 | 10.5 | 0.5 | 1.5 | 0.1 | 1.7 | 0.1 | 15.6 | 1.3 | | |
| FA G2 | 96 | 68 | 2 | 124 | 4 | 21.0 | 0.6 | 92.6 | 36 | 18.5 | 0.8 | 4.6 | 0.2 | 23.3 | 1.1 | 3.3 | 0.1 | 21 | 0.8 | 4.5 | 0.2 | 13.4 | 0.4 | 1.7 | 0.1 | 10.5 | 8.7 | 1.5 | 0.1 | 1.2 | 0.4 | 12.6 | 3.6 | | |

Column groups: light REE (La, Ce, Pr, Nd); medium REE (Sm, Eu, Gd, Tb, Dy); heavy REE (Ho, Er, Tm, Yb, Lu); Thorium (232Th, 230Th); Protactinium (231Pa).

Tab. ES4: Major elements in the sediment traps

| Sample | sampling period | Particulate mass flux | POC fluxe | total Al flux | total Fe flux | BSi fluxe | BioFe flux | Delta Fe | BioAlFlux | deltaAl | fraction of seeded Al in the trap |
|---|---|---|---|---|---|---|---|---|---|---|---|
| - | day | mg/m2/d | mg/m2/d | mg/m2/d | mg/m2/d | mg/m2/d | mg/m2/d | nmol/L | mg/m2/d | nmol/L | |
| Tyr C1 | 3 | 0.8 | 0.3 | ND | ND | ND | 0.0001 | 0.01 | | | |
| Tyr C2 | 3 | 1.5 | 0.5 | ND | ND | ND | 0.0003 | 0.02 | | | |
| Tyr D1 | 3 | 1 704 | 21.1 | 85 | 45 | 31 | 0.0098 | 0.63 | 0.103 | 13.7 | 62% |
| Tyr D2 | 3 | 1 652 | 22.7 | 78 | 42 | 34 | 0.0106 | 0.68 | 0.113 | 15.1 | 57% |
| Tyr G1 | 3 | 1 841 | 24.4 | 87 | 48 | 40 | 0.0113 | 0.73 | 0.136 | 18.1 | 64% |
| Tyr G2 | 3 | 1 805 | 27.2 | 89 | 47 | 41 | 0.0126 | 0.82 | 0.139 | 18.5 | 65% |
| | | | | | | | | | | | |
| Ion C1 | 3 | 2.0 | 0.3 | ND | ND | ND | 0.0001 | 0.01 | | | - |
| Ion C2 | 3 | 0.8 | 0.4 | ND | ND | ND | 0.0002 | 0.01 | | | - |
| Ion D1 | 3 | 1 680 | 22.4 | 81 | 44 | 29 | 0.0104 | 0.67 | 0.097 | 13.0 | 59% |
| Ion D2 | 3 | 756 | 10.8 | 37 | 20 | 15 | 0.0050 | 0.32 | 0.052 | 6.9 | 27% |
| Ion G1 | 3 | 1 349 | 19.0 | 66 | 35 | 27 | 0.0089 | 0.57 | 0.091 | 12.2 | 48% |
| Ion G2 | 3 | 1 257 | 17.5 | 59 | 32 | 20 | 0.0081 | 0.52 | 0.067 | 8.9 | 43% |
| | | | | | | | | | | | |
| Fast C1 | 4 | 0.5 | 0.1 | ND | ND | ND | 0.0001 | 0.01 | | | |
| Fast C2 | 4 | 1.0 | 0.2 | ND | ND | ND | 0.0001 | 0.01 | | | |
| Fast D1 | 4 | 758 | 9.7 | 36 | 19 | 10 | 0.0045 | 0.39 | 0.035 | 6.2 | 35% |
| Fast D2 | 4 | 881 | 11.9 | 42 | 23 | 12 | 0.0056 | 0.48 | 0.040 | 7.2 | 41% |
| Fast G1 | 4 | 684 | 10.3 | 33 | 18 | 10 | 0.0048 | 0.41 | 0.033 | 6.0 | 32% |
| Fast G2 | 4 | 628 | 9.3 | 30 | 16 | 10 | 0.0043 | 0.37 | 0.035 | 6.2 | 29% |

5    bio-Fe-flux calculated based on a Fe/C ratio of 100 µmol/mol. bio-Al-flux calculated based on a Al/Si ratio of 8000 µmol/mol.

Tab. ES5: Release ratio of trace elements relative to thorium (mol/mol). [b29]

| | Fe/Th | Al/Th | error | La/Th | error | Ce/Th | error | Pr/Th | error | Nd/Th | error | Sm/Th | error | Eu/Th | error | Gd/Th | error | Tb/Th | error | Dy/Th | error | Ho/Th | error | Er/Th | error | Tm/Th | error | Yb/Th | error | Lu/Th | error | Pa/Th |
|---|---|---|---|---|---|---|---|---|---|---|---|---|---|---|---|---|---|---|---|---|---|---|---|---|---|---|---|---|---|---|---|---|
| ION_D | <200 | 17000 | 3000 | 13 | 3 | 29 | 6 | 4.4 | 0.8 | 20 | 4 | 4.5 | 0.8 | 1.0 | 0.2 | 4.5 | 0.8 | 0.6 | 0.1 | 3.7 | 0.7 | 0.7 | 0.1 | 1.8 | 0.3 | 0.2 | 0.0 | 1.3 | 0.2 | 0.18 | 0.03 | <0.0001 |
| ION_G | <874 | 28000 | 3400 | 23 | 3 | 50 | 5 | 7.1 | 0.7 | 32 | 3 | 6.8 | 0.7 | 1.5 | 0.1 | 7.1 | 0.7 | 1.0 | 0.1 | 5.5 | 0.5 | 1.0 | 0.1 | 2.6 | 0.3 | 0.3 | 0.0 | 1.9 | 0.2 | 0.25 | 0.03 | |
| FAST_D | <1515 | 17200 | 1400 | 10 | 1 | 26 | 3 | 3.9 | 0.4 | 17 | 2 | 3.9 | 0.4 | 0.9 | 0.1 | 5.0 | 0.6 | 0.6 | 0.1 | 3.4 | 0.4 | 0.6 | 0.1 | 1.6 | 0.2 | 0.2 | 0.0 | 1.1 | 0.1 | 0.16 | 0.02 | |
| FAST_G | <717 | 23000 | 2800 | 17 | 4 | 38 | 7 | 5.4 | 0.9 | 24 | 4 | 5.2 | 0.7 | 1.1 | 0.2 | 5.1 | 0.7 | 0.7 | 0.1 | 4.2 | 0.5 | 0.8 | 0.1 | 2.0 | 0.3 | 0.2 | 0.0 | 1.4 | 0.2 | 0.17 | 0.02 | |

[Figure]

Fig. ES1: Transect of the PEACETIME cruise. 10 short stations are numbered from St.1 to St.10. Stars named TYR, ION, and FAST indicate the 3 long stations where tank experiments were conducted.

[Figure]

Fig. ES2: Shale-normalized concentrations of filtered seawater and trapped particles. Note the scale break in the middle of the graph.

[Figure]

Fig. ES3: $^{230}$Th/$^{232}$Th ratio during the tank experiments. a) ION station. b) FAST station. Crosses correspond to samples collected before dust addition (t = 0).[b30]

.

---

## Author Response (AR2)

Gif sur Yvette, February 3rd, 2021

Dear Editor,

Please, find enclosed the revised version of manuscript bg-2020-247 "Contrasted release of insoluble elements (Fe, Al, REE, Th, Pa)after dust deposition in seawater: a tank experiment approach" by Matthieu Roy-Barman, Lorna Folio, Eric Douville, Nathalie Leblond, Fréderic Gazeau, Matthieu Bressac, Thibaut Wagener, Céline Ridame, Karine Desboeufs, and Cécile Guieu.

We greatly appreciated detailed review and copy editing of MS by C. Klaas. We have followed it as closely as a possible. A detailed response to the main questions raised by Christin Klaas is given below.

Yours faithfully,

Matthieu Roy-Barman
* * *
[CK]    Dear author,

The experiments and results presented in this manuscript are novel and of great interest. I do have some issues that I believe should be addressed before publication.

The text needs improvements (English). Suggestions are given in the annotated manuscript.

**[MRB] All the suggested text improvement have been included in the ms.**

[CK] The authors often report approximate (~) values for quantities, even though data is available. Please in these cases refer to the exact values and their uncertainties.

**[MRB] All the ~ symbols have been removed and changed for exact values and uncertainties or for the value range.**

[CK] I failed to understand how the solubility of tracers (section 4.1) was estimated for several reasons:

1) the terms in the equations, in particular Eq.1, are not defined in the text, and the explanations in the text are unclear (see also comments in the annotated manuscript).

**[MRB] All the terms used in equation 1 are now defined. Below I group and answer to the main questions reported in the annoted ms on section 4.1:**

Page 8 line 16 :

[CK] *Unclear. Do you mean you use the the difference between initial and maximum dissolved concentration from the treatment that gave maximum dissolution? or do you take the difference bewteen treatment with maximum dissolution and the control tank?*

**[MRB] We use the data obtained at t = 1 h and we calculate the difference between the element concentration in the D or G experiment averaged over tank replicates 1 and 2 and the concentration in the C tanks also averaged over tank replicates 1 and 2.**

Page 8 line 21 // line 25 // line 26

[CK] *why not use the mass of dust added minus the amount that sedimented? // this is not a robust explanation since the % dissolution also depend on the differences in concentration (nominator) in Eq. 1. Again, why not use the input-trap material? // How is this estimated? If you can estimate this, why not use the mass of dust added (or for REE, 232Th and Pa the particulate amount) minus the amount that sedimented in the denominator of Eq.1? Where are the analysis of particulate suspended material and sedimented material that would allow you to close the budgets?*

**[MRB] We calculate the dissolved fraction relative to the total mass of dust added to the tank (m) rather than to the mass of dust remaining in the tank at the end of the experiment (= dust added minus the amount that settled in the trap) for several reasons:**

**- First, we are interested in relating amount of element released to the total flux of dust deposited at the sea surface, whether the dust particles sink rapidly or not. We are interested in the bulk effect of dust deposition.**

**- the rationale for using the mass of dust remaining in suspension at the end of the experiment is that particles remaining in suspension are more likely to release dissolved elements in seawater. However, as elements are released very rapidly (at least for Th and REE) and as we do not know if dust sinks in the trap at the beginning of the experiment, at the end of the experiment or more or less continuously, it is unclear if particles remaining in suspension had a chance to release more elements than particles setting in the trap.**

**- To test the effect of particle dynamics within the tanks (settling or remaining in suspension), we compare the results obtained for ION D1 and ION D2 for which the fraction of particles remaining in suspension (= input to the tank – output in the trap) varies almost by a factor of 3 (25% of m for ION D1 and 68% of m for ION D2). Despite different dust fractions remaining is suspension, the quantity of element released in solution evolves similarly during the ION D1 and ION D2 experiments (dissolved Al, REE, Th concentrations identical within analytical uncertainties).**

Page 8 line 22 // line 25:

[CK] *How so? You assume that their concentration in plankton and other suspended particles is the same than in sedimenting material? I have difficulties in believing this assumption. // Again as for REE and Th, this applies only if Pa in plankton and other suspended material in the tank is the same as in the trap. That seems like a big IF no?*

**[MRB] We assume that :**

- the concentrations of insoluble elements of the suspended dusts are identical to the concentrations in the trapped particles.

- the quantity of insoluble elements newly produced biogenic particles or plankton are negligible compared to the quantity of insoluble elements of the dusts. This is both because we expect low concentrations in the biogenic particles and because the mass of biogenic particles is much lower than the mass of dust given the mass of dust (3.6 g used in each tank).

Page 8 line 23 :

 [CK] *of what? Mass flux?*

**[MRB] concentration**

[CK] *Since carbonates are part of the natural dust input, what is your point in correcting for that?*

**We have removed the correction of the concentrations for carbonate dissolution**

**[MRB] We have removed the correction of the concentrations for carbonate dissolution.**

Page 8 line 31:

 [CK] *Why do you discuss THE outlier here and not the other treatments? What is the point of this paragraph?*

**[MRB] Unfortunately, there is not only one outlayer and it is difficult and risky to draw any trend on Fe dissolution from these tank experiments. Therefore, we limit the discussion/interpretation to the calculation of an upper limit for Fe dissolution.**

[CK] 2) For the elements that were not measured in the original dust, the authors use the values from the sedimented material. Here the methods and assumptions are also poorly described.

**[MRB] The assumptions for using the REE, Th and Pa concentrations in the trapped material as an analog for the original dust are:**

**- REE, Th and Pa concentrations were identical in the sedimented material and in the suspended particles;**

**- for REE, Th and Pa, the contribution of plankton and other biogenic material produced during the experiment was negligible in the sedimented material given the high dust load recovered in the trap**

**- carbonate dissolution (see section 3.6) add negligible effect.**

**These hypotheses have been added to the main text.**

[CK] In short: section 4.1 needs some clarifications, and the assumptions should be clear and also discussed.

**[MRB] Done above**

[CK] Further minor aspects that need clarification are commented in the combined annotated manuscript and supplement.

**[MRB] Answer to these questions have been included in the text of the ms and of the supplement.**

**[MRB] The only point for which we did not follow CK suggestion concerns TEP (page 11, line 18): we agree that other particles or colloids mat scavenge Th. However, many studies highlight the putative role of TEP. Therefore, we think it is worth mentioning that during the present experiments that TEP do not seem to be the main Th scavenger.**

---

## Editor Decision (ED2)

[revised manuscript text omitted]

 Fig 1

[Figure]

Fig 2a.

[Figure]

15    Fig 2b.

[Figure]

Fig. 3

[Figure]

15    Fig. 4:

[Figure]

Fig. 5.

[Figure]

Fig. 6

---

## Author Response (AR3)

Gif sur Yvette, March 4th, 2021

Dear Editor,

Please, find enclosed the revised version of manuscript bg-2020-247 "Contrasted release of insoluble elements (Fe, Al, REE, Th, Pa)after dust deposition in seawater: a tank experiment approach" by Matthieu Roy-Barman, Lorna Folio, Eric Douville, Nathalie Leblond, Fréderic Gazeau, Matthieu Bressac, Thibaut Wagener, Céline Ridame, Karine Desboeufs, and Cécile Guieu.

We greatly appreciated detailed copy editing of MS by C. Klaas. We have applied all changes suggested by Christin Klaas. We have also corrected some remaining typos.

Yours faithfully,

Matthieu Roy-Barman